



**Modeling Seismic Hazard and Landslide Potentials in Northwestern Yunnan,**
**China: Exploring Complex Fault Systems with multi-segment rupturing in a**
**Block Rotational Tectonic Zone**
Jia Cheng[1*], Chong Xu[2], Xiwei Xu[1], Shimin Zhang[2], Pengyu Zhu[2]
1.  School of Earth Science and Resources, China University of Geosciences (Beijing),
Beijing, 100083, China
2.  National Institute of Natural Hazards, Ministry of Emergency Management of China,
Beijing, 100085, China
*Corresponding author:
Jia Cheng (jiacheng@cugb.edu.cn, jiacheng@gmail.com)
Address: China University of Geosciences, No. 29, Xueyuan Road, Haidian, Beijing, 100083,
China
Phone: +86-10-13466515670
Fax: +86-10-82322264
**Abstract**
The Northwestern Yunnan Region (NWYR), located on the southeastern edge of the
Tibetan Plateau, is characterized by a combination of low-crustal flow and gravitational
collapse, giving rise to a complex network of active faults. This presents significant
seismic hazards, particularly due to the potential for multi-segment ruptures and
resulting landslides, as demonstrated by the historical 1515 $M$7.8 Yongshen Earthquake.
This article presented a novel seismic hazard modeling study for the NWYR,





integrating fault slip parameters and assessing multi-segment rupturing risks. Among
the four potential multi-segment rupture combination models examined, Model 1,
characterized by multi-segment rupture combinations on single faults, particularly
fracturing the Zhongdian fault, emerges as the most suitable for the NWYR, supported
by the alignment of modeled seismicity rates with fault slip rates. Our analysis
demonstrated that peak ground-motion acceleration values, calculated with a 475-year
return period from modeled seismicity rates, exhibited a strong correlation with fault
distribution, averagely higher than the China Seismic Ground Motion Parameters
Zonation Map. Furthermore, we conducted simulations to forecast landslide occurrence
probabilities across our peak ground-motion acceleration distribution map. Our
findings underscored that the observed combinations of multi-segment ruptures and
their associated behaviors were in alignment with the small block rotation triggered by
the gravitational collapse of the Tibetan Plateau. This result highlighted the intricate
interplay between multi-segment rupturing hazards and regional geological dynamics.
**Key Words:**
Northwestern Yunnan Region; multi-segment rupture; probability seismic hazard
analysis; landslide probabilities
**1.  Introduction**
The collision of the Eurasia Platea and the Indian plate makes the Tibetan Plateau world
highest altitude of 4000+ m averagely. The eastern extrusion of the crust in the Tibetan


Plateau, associated with the wedged Eastern Himalayan syntaxis, initiates a clockwise
rotation of crustal deformation in the southeastern margin of the Tibetan Plateau (Figure
1) (Zhang et al., 2004; Gan et al., 2007; Wang and Shen, 2020). The Northwestern
Yunnan Region (NWYR), in the west part of the southeastern margin of the Tibetan
Plateau, borders the Tibetan Plateau, with the Lijiang-Xiaojiang fault serving as a
boundary fault that separates the Tibetan Plateau, boasting an average altitude of over
3000 meters, from the Yunnan Region, which maintains an average altitude of over
2000 meters (Yu et al., 2022; Zhang et al., 2022) (see Figure 1). Unlike thrust faults in
the plateau boundary, such as the Longmenshan fault ruptured by the 2008 $M_W$7.9
Wenchuan earthquake, the Holocene slip type of the Lijiang-Xiaojinhe fault is sinistral,
with a strike-slip rate of approximately 3 mm/yr, as determined by geological (Xu et al.,
2003; Shen et al., 2005; Ding et al., 2018; Gao et al., 2019) and geodetic data (Gan et
al., 2007; Cheng et al., 2012). The peculiar slip behavior of the Lijiang-Xiaojinhe fault
has garnered considerable attention in studies pertaining to crustal structure, fault
activities, and earthquake hazards (Xu et al., 2003; Cheng et al., 2012; Zhao et al., 2013;
Bao et al., 2015; Zhang et al., 2020; Huang et al., 2022; Zhang et al., 2022; Dai et al.,
2023). Zhang et al. (2020), a shear-wave velocity model was employed to reveal that
three faults - the Longmenshan fault, the Lijiang-Xiaojinhe fault, and the Chenghai fault
- delineate a low-velocity belt. This investigation unveiled the presence of low-crustal
flow beneath the Northwestern Yunnan Region (NWYR). Similarly, Zhang et al. (2022)
utilized magnetotelluric (MT) observations in the southern vicinity of the Lijiang-
Xiaojinhe fault, corroborating these findings and emphasizing the NWYR as a pathway



for ductile low-crustal flow. Analogously, a GPS study by Cheng et al. (2012) yielded
comparable results. Upon eliminating the rigid rotation component from the regional
GPS velocity field, they demonstrated a clockwise rotation propelled by ductile crustal
flow, particularly accelerated within the NWYR. They posited that this acceleration in
clockwise rotation might also be intensified by the tensional drag originating from the
Burma Plate.
The intricate network of crustal deformation encompassing the Northwestern Yunnan
Region (NWYR) introduces complexity to the slip behavior of faults and the focal
mechanisms of recent earthquakes. Within this area, three distinct fault slip behaviors
are observed: the NE-trending Lijiang-Xiaojinhe fault displays left-lateral strike-slip,
NW-trending faults exhibit right-lateral strike-slip, and North-South trending faults
demonstrate normal slip (see Figure 2). The presence of faults with diverse rupture
behaviors contributes to the complexity of earthquake hazards. Historically, these faults
have been associated with significant seismic events and numerous casualties. Notably,
three earthquakes with $M7+$ have occurred in the NWYR: the Yongsheng earthquake
of 1515 ($\sim M7.5$) on the Chenghai fault, the Midu earthquake of 1652 ($\sim M7$) on the Red
River fault, and the Dali earthquake of 1925 ($\sim M7$) on the Diancangshan East fault.
Additionally, the 1990 Lijiang earthquake ($M_S7.0/M_W6.6$) occurred on the Yulong East
fault, exhibiting dominant normal slip behavior. Historical and paleo-earthquake
studies suggest that nearly all of these faults have the potential to generate catastrophic
earthquakes (e.g., Ding et al., 2018; Ren et al., 2007; Chang et al., 2014), and induced
numerous landslides (Institute of Geology-State Seismological Bureau, and Yunnan



Seismological Bureau, 1990; Huang et al. 2021).
Fieldwork studies and focal mechanisms of recent earthquakes underscore the
complexity of fault slip behaviors in this tectonic environment (Figure 2). Both
historical and instrumental earthquakes have affected nearly all faults in the region,
emphasizing the seismic risks in NWYR. For instance, the 2013 Deqin earthquake
swarm, reaching a maximum magnitude of $M_S5.9$/$M_W5.7$ on August 31 (Wu et al.,
2015), and the 2021 Yangbi earthquake swarm, reaching a maximum magnitude of
$M_S5.9$/$M_W6.1$ on May 21 (Zhou et al., 2022), are noteworthy seismic events (refer to
Figure 2). The 2013 $M_W5.7$ Deqin earthquake swarm, characterized by tensional stress,
occurred at the intersection of the Zhongdian fault and the southern part of the
Jinshajiang fault, illustrating the susceptibility of the regional stress field to disturbance.
Conversely, the 2021 $M_W6.1$ Yangbi earthquake swarm occurred at the connection point
of the dominant dextral strike-slip faults, namely the Red River fault and the Weiqi-
Qiaohou fault, representing a different tectonic environment compared to the 2013
$M_W5.7$ Deqin earthquake swarm. This distinct setting suggests that either of these two
faults may be at risk of seismic activity during the pre-earthquake period.
Due to the high altitude, dense vegetation, and easily weathered conditions, obtaining
accurate fault slip rates poses a significant challenge, often resulting in notable errors.
Recent studies have provided fresh insights into slip rates and fault behaviors, offering
the potential to enhance the precision of seismic hazard models. For instance,
determining the dextral slip rate of the Zhongdian fault has proven particularly difficult
due to high error margins. Recent research has evaluated the Holocene dextral slip rate


to be ~1.5±0.2 mm/yr based on displacements of water-ice remains (Wu et al., 2019),
and ~2.1±0.2 mm/yr based on displacements of river terraces (Chang et al., 2014), both
utilizing Optically Stimulated Luminescence (OSL) dating. These values notably differ
from the right-lateral slip rate of 4~6 mm/yr estimated by Shen et al. (2001) based on
gully displacements from the last glacial period, but are more aligned with the rate
derived from GPS velocity data (Cheng et al., 2012). Incorporating these updated fault
slip rates into regional seismic hazard models holds the promise of enhancing their
accuracy. Therefore, integrating these new slip rates into the regional seismic hazard
model is crucial to ensure the reliability of the results.
Given the inherent challenges of fieldwork studies on fault activities, only a limited
number of investigations have been conducted regarding seismic hazard analysis in the
Northwest Yunnan region. Among these studies, Zhou et al. (2004) conducted a micro-
zonation of seismic hazards in the NWYR. They examined regional fault activities
through field surveys and estimated the potential maximum magnitude of these faults.
Their approach involved outlining polygons around the source faults to divide them
into different potential seismic sources and calculating historical seismicity rates within
these polygons. This methodology is widely employed in seismic hazard modeling in
China, particularly in the national seismic hazard map of the China Seismic Ground
Motion Parameters Zonation Map (CSGMPZM) (Gao et al., 2015). CSGMPZM also
utilized this methodology to assess potential maximum magnitudes and compute
seismicity rates. However, their studies often did not integrate fault geometry models,
especially fault segmentation models. Consequently, the fault geometry, including


rupture length and area, may not be accurately linked to the magnitude of large
earthquakes. Furthermore, it is crucial to recognize the potential occurrence of multi-
segment rupturing, which has not been documented in historical records. Similarly,
seismicity rates were typically derived solely from historical earthquakes and were not
synchronized with fault slip rates. Relying solely on historical earthquakes for
seismicity rate calculations may lead to either overestimation or underestimation of
seismic hazards.
In this article, we developed a regional seismic hazard model for the Northwestern
Yunnan Region (NWYR), accounting for fault slip behaviors, the potential occurrence
of large earthquakes, and the likelihood of multi-segment ruptures. We initially
developed fault segmentation models for the primary active faults in the Northwestern
Yunnan Region (NWYR), drawing on recent geological research on fault segmentation
and geological fault slip rates. Subsequently, we employed the SHERIFS code (Chartier
et al., 2017; 2019) to simulate seismicity rates across possible multi-segment
combination models. We identified the multi-segment combination model that best
aligns with the majority of fault slip rates, considering fault segmentation and historical
seismicity rates. Ultimately, we calculated the Peak Ground Acceleration (PGA) with a
10% probability of exceedance within 50 years using the seismicity rates from the
selected fault segmentation models. The exploration of multi-segment rupture
combinations, along with the resultant modeled seismicity rates and PGA values, offers
valuable insights into the seismic hazard present in the NWYR. Leveraging the modeled
PGA values, we employed a machine learning model to compute the probability



distribution of landslides induced by potential seismic hazards. This increased precision

and reliability will be invaluable for guiding disaster preparedness initiatives, land-use

planning, and infrastructure resilience strategies in the area.

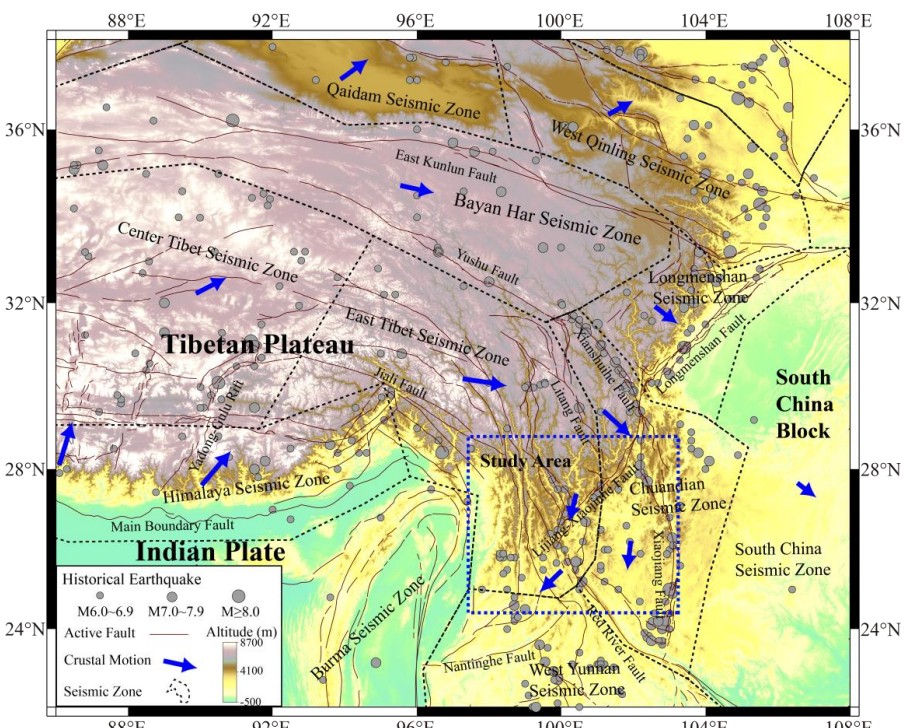

Figure 1. Tectonic environment of the Eastern Tibetan Plateau and the location of the

Northwestern Yunnan Region (NWYR). The dashed rectangle delineates the study area,

while dashed polygons depict the seismic zones delineated by Rong et al. (2020).

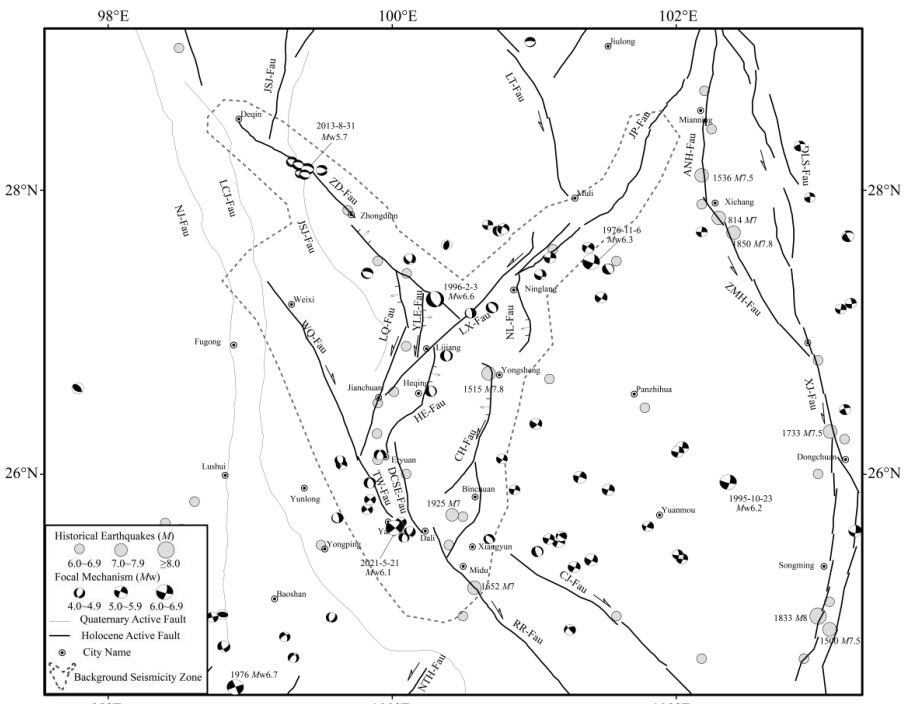

Figure 2. Regional active faults and historical earthquake activities in the NWYR. The focal mechanisms of recent earthquakes (1976~2023) are sourced from the global centroid moment tensor (GCMT) catalog. Earthquakes with $M$6+ are sourced from the moment magnitude ($M_W$) catalog of Cheng et al. (2017). The dashed line represents the division for background seismicity calculation, which extends 20 km from the faults. JP-Fau: Jinping fault; LT-Fau: Litang fault; ANH-Fau: Anninghe fault; ZMH-Fau: Zemuhe fault; XJ-Fau: Xiaojiang fault; CJ-Fau: Chuxiong-Jianshui fault; RR-Fau: Red River fault; NTH-Fau: Nantinghe fault; DYJ-Fau: Dayingjiang fault; NJ-Fau: Nujiang fault; LCJ-Fau: Lancangjiang fault; JSJ-Fau: Jinshajiang fault; ZD-Fau: Zhongdian fault; LX-Fau: Lijiang-Xiaojinhe fault; WQ-Fau: Weixi-Qiaohou fault; YLE-Fau: Yulong East fault; LQ-Fau: Longpan-Qiaohou fault; HE-Fau: Heqing-Eryuan fault; CH-Fau: Chenghai Fault; DCSE-Fau: Diancangshan East Fault; TW-Fau: Tongdian-Weishan Fault.


**2. Fault slip rates, Segmentation, and Multi-segment Rupture Combinations**
**Models**
**2.1 Fault slip rates, Segmentation**
In the NWYR, the Lijiang-Xiaojinhe fault, characterized by its left-lateral strike-slip
rate, and the northern segment of the Red River fault, which displays significant right-
lateral strike-slip movement, play pivotal roles in crustal deformation. Moreover, as a
result of the southward extrusion of the Tibetan Plateau, NE-trending faults such as the
Lijiang-Xiaojinhe fault also manifest a left-lateral strike-slip component. These
observations underscore the complex interplay of fault dynamics in the NWYR, as
elucidated by previous studies (Gan et al., 2007; Cheng et al., 2012; Wang and Shen,

191 2020).

To counterbalance the southwestward crustal extrusion (Wang et al., 1998; Cheng et al.,
2012), several other faults in the region, such as the Chenghai fault, the Ninglang fault,
the Heqing-Eryuan fault, the Yulong East fault, and the Longpan-Qiaohou fault (also
known as the Jianchuan fault), also exhibit a component of normal slip rate as well
(Institute of Geology-State Seismological Bureau, and Yunnan Seismological Bureau,
1990; Han et al., 2004). In contrast, the Zhongdian fault and the northern part of the
Red River fault, including the Weixi-Qiaohou fault and the Diancangshan East fault,
exhibit right-lateral strike-slip movement (Zhou et al., 2004; Han et al., 2005). Recent
focal mechanisms of intermediate earthquakes indicate a complex regional stress field,
featuring both strike-slip and normal faulting regimes (Figure 2). Table S1 provides an
overview of the observed fault slip rates in the NWYR.



The Lijiang-Xiaojinhe fault serves as a boundary fault delineating the Tibetan Plateau
from the Central Yunnan block (Xu et al., 2003; Cheng et al., 2012). We divided the
Lijiang-Xiaojinhe fault into 10 segments (the F1~F10 segments in Figure 3) based on
fault geometry and its intersection with other faults. For the F1 segment, known as the
Jinpingshan fault, recent fault mapping reveals a Holocene left-lateral slip rate ranging
from 1.3~2.7 mm/yr derived from gully displacement across the segment, while the
vertical slip rate is approximately 0.2 mm/yr (Mr. Rui Ding, 2024, private
communication).
Regarding the F5~F10 segments, Gao et al. (2019) demonstrated that the Hongxing-
Jianshanying segment (F6 segment in Figure 3) exhibited a Holocene left-lateral slip
rate of 3.32±0.22 mm/yr with a normal slip rate component of 0.35±0.02 mm/yr,
whereas the Runan-Nanxi segment (F10 segment in Figure 3) had a Holocene left-
lateral slip rate of 2.37±0.20 mm/yr. Accordingly, we applied the slip rate of the F6
segment for the F4~F7 segments and the slip rate of the F10 segment for the F8~F10
segments. Notably, we considered the strike-slip motion of the F5~F10 segments to
originate from two sources: the strike-slip Jinpingshan fault and the strike-slip of the
Litang fault, aligning with the observed clockwise rotation of regional crustal
deformation around the Litang fault and the Lijiang-Xiaojinhe fault. Consequently, we
inferred the left-lateral strike-slip rate of the F4 segment to be ~2.1 mm/yr, consistent
with the southern section of the Litang fault (Zhou et al., 2007). However, the F2 and
F3 segments, which link the F1 and F4 segments, lack recorded fault slip rates from
fieldwork studies. In this regard, we assigned a conservative estimate, employing half



the value of the strike-slip rate of the F1 segment for both the F2 and F3 segments,
approximately 1.2 mm/yr.
For the Longpan-Qiaohou fault (comprising the F11~F14 segments), we delineated it
into four distinct segments based on the fault mapping data provided by Wu et al. (2023).
The sinistral slip rate of the Longpan-Qiaohou fault was estimated at ~2.2 mm/yr over
the past 3500 years, with a normal slip rate of 0.23 mm/yr (Institute of Geology-State
Seismological Bureau, and Yunnan Seismological Bureau, 1990).
As for the Yulong East fault, we segmented the fault into two segments, namely the F15
and F16 segments, utilizing fault mapping data and Quaternary sedimentary
distribution. The slip rate of the Yulong East fault was assessed by Han et al. (2005),
who determined that the Quaternary left-lateral and normal slip rates are 0.84 mm/yr
and 0.70 mm/yr, respectively, derived from the displacement observed in a gully
crossing the fault.
Regarding the Zhongdian fault, we partitioned it into six segments, designated as the
F17~F22 segments, based on fault mapping data (Wu et al., 2023). The Holocene
dextral slip rate of the Zhongdian fault is estimated to be approximately 1.7-2.0 mm/yr,
with a minor normal slip rate of 0.6-0.7 mm/yr based on terrace displacement across
the fault (Chang et al., 2014).
For the Heqing-Eryuan fault, we segmented it into two sections, labeled as the F23 and
F24 segments. The Quaternary dextral slip rate and normal slip rate of the Heqing-
Eryuan fault were reported to be around 2 mm/yr and 0.7~1.0 mm/yr, respectively, as



documented by the Institute of Geology-State Seismological Bureau, and Yunnan
Seismological Bureau (1990). Additionally, recent research by Sun et al. (2017) yielded
similar fault slip rate results, indicating a left-lateral slip rate of 1.80 mm/yr and a
vertical slip rate of 0.28 mm/yr since the Pleistocene.
The Ninglang fault is primarily characterized as a left-lateral strike-slip fault, although
it exhibits a minor normal slip component of less than 0.1 mm/yr at the basin margin.
The strike-slip rate of the Ninglang fault, as determined from fault mapping work
conducted by Dr. Panxing Yang from Institute of Earthquake Forecasting, China
Earthquake Administration (private communication), was estimated to be less than 1
mm/yr. For our analysis, we opted to utilize a median value of 0.5±0.4 mm/yr for the
strike-slip rate of the Ninglang fault. Based on the distribution of Quaternary sediments,
we divided the Ninglang fault into two distinct segments, designated as the F25 and
F26 segments.
For the Chenghai fault, the sinistral slip rate has been estimated to range from 2.5 to
3.0 mm/yr, determined from the erosion rate of the Jinshajiang River crossing the fault.
Additionally, the normal slip rate is reported to be between 0.7 and 1.0 mm/yr, assessed
from the lift rate of the fault scarps (Institute of Geology-State Seismological Bureau,
and Yunnan Seismological Bureau, 1990), which is consistent with the findings of Tang
et al. (2017). We divided the Chenghai fault into three segments, i.e., the Chenghai
segment (the F27 segment), the Qina segment (the F28 segment), and the Bingchuan
segment (the F29 segment), based on the sedimentary distribution (Huang et al., 2018;
Yu et al., 2005).





The southern end of the Longpan-Qiaohou fault separates the Weixi-Qiaohou fault from
the Tongdian-Weishan fault. We segmented these two faults into six segments each
based on fault mapping data and Quaternary sedimentary distribution. Concerning the
slip rate of the Tongdian-Weishan fault, the dextral slip rate in the Late Pleistocene is
estimated to be ~1.8-2.4 mm/yr, with a normal slip rate of 0.17-0.35 mm/yr, calculated
from the displacement of fault scarps (Chang et al., 2016). In contrast, for the Weixi-
Qiaohou fault, the dextral slip rate is ~1.25 mm/yr, while the normal slip rate is ~0.91
mm/yr since the Late Pleistocene (Ren et al., 2007). Comparing these rates to the dextral
slip rate from the middle section of the Red River fault, which is reported to be 1.1 ±
0.4 mm/yr (Shi et al., 2018), it is evident that the dextral slip rates decrease from the
northwest to the southeast across the Red River fault system, encompassing the Weixi-
Qiaohou fault, the Tongdian-Weishan fault, and the Red River fault.
The Diancangshan East Fault is the seismogenic fault of the 1925 $M$7 Dali earthquake.
We deduced that the Diancangshan East fault is a dominant normal slip fault as the
boundary fault of the Dali basin and the Erhai Lake. The normal slip rate of this fault is
1-2 mm/yr (Guo et al., 1984; Zhou et al., 2004).
Additionally, we incorporated the F37 and F38 segments of the northern part of the Red
River fault into our segmentation model. The right-lateral strike-slip rate of these two
segments is ~1.1 mm/yr. Figure 3 illustrated the segmentation model of the faults in the
NWYR.



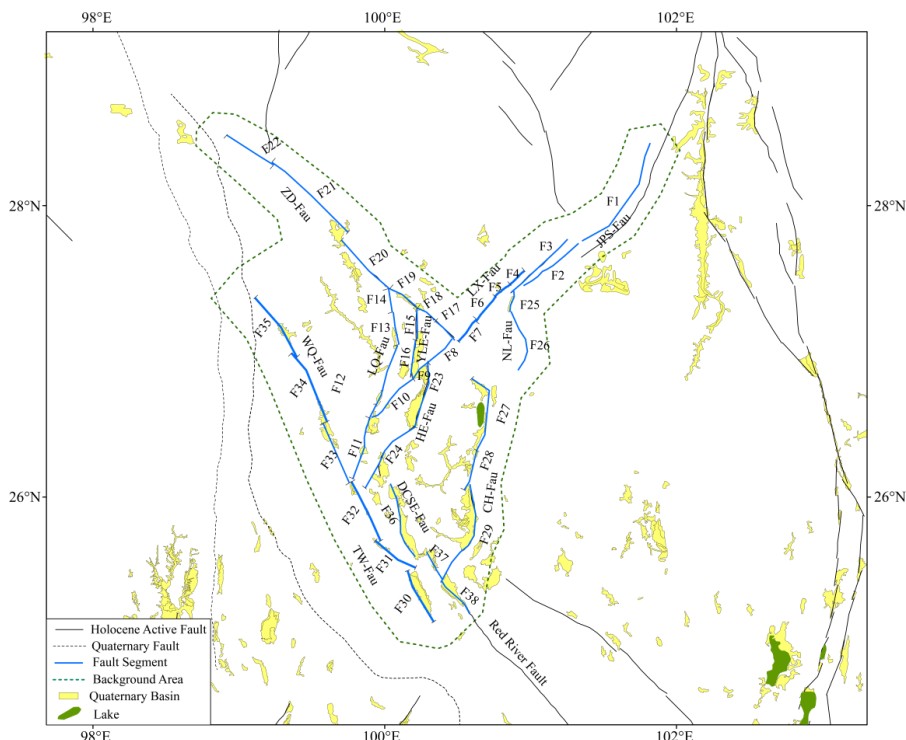

Figure 3. Fault segmentation model for the Northwestern Yunnan Region (NWYR).

In which, the Quaternary Basin distribution were from Deng et al. (2003); the fault

data are from Wu et al. (2023).

## 2.2 Multi-segment Rupture Combinations Models

Based on the segmentation model, fault rupture behaviors, and the intersections

among fault segments, we devised four multi-segment rupture combination models for

the fault segments in the NWYR (Figure 4). The $M_W$6.6 Lijiang earthquake on February

3, 1996, represents a significant normal rupture event that occurred on the Yulong East

fault. This earthquake stands out as the most substantial seismic event in the NWYR

since the 1970s, underscoring the normal slip behavior of the Yulong East fault. This





observation suggests potential implications for the rupture behavior of the Zhongdian
fault.
In Model 1, we exclusively examined the multi-segment rupture combinations
within the same faults. Specifically, for the Zhongdian fault, we integrated the multi-
segment rupturing of the F17 and F18 segments, as well as the multi-segment rupturing
of the F19 and F20 segments. This approach considered the normal slip behavior of the
Yulong East fault (F15 and F16 segments) and its potential impact on Quaternary
sedimentary distribution between the F18 and F19 segments of the Zhongdian fault.
Subsequently, in Model 2, we evaluated the plausibility of multi-segment rupturing
occurring across the F17~F20 segments.
In the NWYR, the prevailing fault behavior is sinistral slip along the northeast-
trending faults, a trend consistent with the observed clockwise rotation in regional
crustal deformation (Cheng et al., 2012) and the presence of ductile low-crust flow
(Zhang et al., 2022). The sinistral slip observed along the Longpan-Qiaohou fault may
hinder the dextral slip occurring along the Weixi-Qiaohou fault, which extends from the
Tongdian-Weishan fault, contributing to the decrease in dextral slip rates observed from
the Weixi-Qiaohou fault to the Tongdian-Weishan fault. In Model 3, we integrated the
multi-segment rupture combination of the Weixi-Qiaohou fault (the F33~F35 segments)
and the Tongdian-Weishan fault (the F30~F32 segments).
In 2023, two earthquakes of $M_W7.8$ and $M_W7.5$ successively ruptured the East
Anatolia fault region in Turkey (Xu et al., 2023; Petersen et al., 2023). The rupture of
the first earthquake, with Mw7.8, initiated on the splay Narli fault and propagated
bilaterally along the main East Anatolia fault (Liu et al., 2023). Consequently, we took
into account the possibility of rupture propagation from one fault to another in our
rupture combinations. Using Model 4, we investigated whether the rupture on the





Lijiang-Xiaojinhe fault could propagate to the Longpan-Qiaohou fault. This
consideration was prompted by similarities in the rupture behavior between the F11
segment of the Longpan-Qiaohou fault and the F10 segment of the Lijiang-Xiaojinhe
fault, along with a minor difference in strike (Table S1).

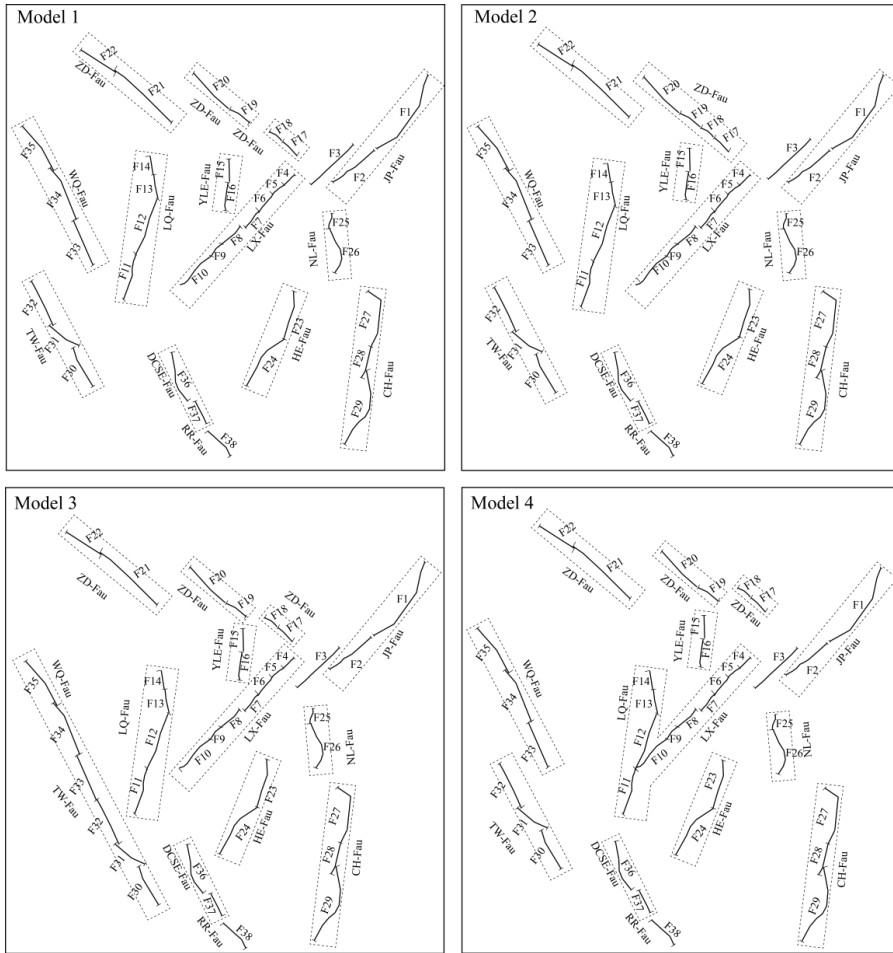


Figure 4 Possible Rupture Combination Models for the Fault Segments in NWYR.
Dashed rectangular show the rupture combinations for each Model.

**3. Multi-segment rupture hazard Modeling**





### 3.1 **Methodology**

In the earthquake hazard modeling, the seismicity rate of the earthquakes should reflect both the fault slip rate and the magnitude-frequency distribution (MFD), e.g., the Gutenberg-Richter (G-R) Relationship (Gutenberg and Richter, 1944) and the Characteristic earthquake model (Schwartz and Coppersmith, 1984). Youngs and Coppersmith (1995) balanced the fault slip rates and the magnitude-frequency relationship in the seismicity rate on the faults. They employed the composite characteristic earthquake model (Y-C) or truncated G-R model to convert the fault slip rate into the seismicity rate on the fault. These converted MFD were widely used in seismic hazard analysis (e.g., Avital et al., 2018; Chartier et al., 2019; Rong et al., 2020). This approach allows for a more comprehensive assessment of earthquake hazards by integrating both fault slip rates and the frequency of seismic events.

For assessing the possibilities and probabilities of multi-segment rupturing, it is essential to represent the seismicity rate of such combinations in the magnitude-frequency relationship for each segment. To achieve this, Chartier et al. (2017; 2019) devised a Python-based code known as SHERIFS. This code employed an iterative process, enabling the balancing of occurrence rates for multi-segment rupturing events alongside intermediate and small earthquakes on each fault segment (Figure 5a). Leveraging historical seismicity data, they utilized the slip rate of each fault segment to convert it into the target MFD, such as the G-R, or the Y-C distribution. This method offered a robust framework for assessing seismic hazard, integrating both single and multi-segment rupture scenarios effectively. Determining the maximum magnitudes of individual fault segments and their combinations could rely on fault length, following rupture scaling laws proposed by researchers like Wells and Coppersmith (1994) and Cheng et al. (2020).



In the final step, they iterated the seismicity rates across magnitude bins associated
with multi-segment rupturing, spanning from large magnitudes down to small
magnitudes, according to the target MFD for each fault segment. However, in many
cases, the fault slip rate or calculated seismicity rates couldn't fully account for the
entire seismic activity. The remaining portion of the fault slip rate for each segment was
attributed to non-main-shock slip (NMS), including processes like post-seismic slip and
silent creep. A non-main-shock (NMS) ratio of ≤30%~40% was typically considered
indicative of a model misfit, potentially due to creeping and specific conditions such as
boundary fault segments or creeping segments (as depicted in Figure 5b). Here, we
adopted a similar approach in simulating seismic hazard modeling for the regional fault
system in the NWYR.

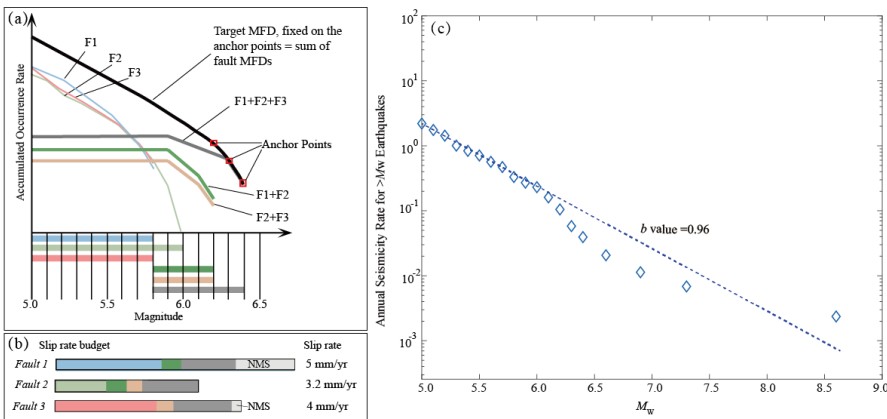

**Figure 5** a. Scheme of the occurrence rate iterative process on the fault segments
constrained by the magnitude-Frequency relationship and fault slip rates;b. Slip
budget of the fault slip rate and its consummation on the earthquakes (Modified from
Chartier et al., 2017);c. Calculated b value for the East Tibet Seismic Zone where
the NWYR located in Figure 1.

Given the fractured structure of the crust in the NWYR, as documented by Cao et





al. (2023), the seismicity distribution in this area was notably complex and differed
significantly from that observed directly on the fault lines. Therefore, in our analysis of
seismicity rates for the whole seismicity rates on the regional faults, we opted to utilize
the Gutenberg-Richter (G-R) relation (Gutenberg and Richter, 1944) as the Magnitude-
Frequency relationship, rather than the Youngs-Coppersmith (Y-C) relation (Youngs
and Coppersmith, 1985).

For estimating the magnitudes based on rupture length, we applied the relationsip

proposed by Cheng et al. (2020) to determine the maximum magnitude for each
individual fault segment as well as their multi-segment combinations. Additionally, we
accounted for a portion of earthquakes with $M$<6.5 as off-fault seismicity. Specifically,
we assigned probabilities of 95%, 90%, 85%, 80%, and 80% for magnitude bins ranging
from 6.0 to 6.4, 5.5 to 5.9, 5.0 to 5.4, 4.5 to 4.9, and 4.0 to 4.4, respectively, based on
prior studies (Chartier et al., 2019; Cheng et al., 2021).

We conducted a calculation of the b-value for the East Tibet Seismic Zone, which

encompasses nearly all of the NWYR, as illustrated in Figure 1. The earthquake catalog
utilized for this analysis was sourced from Cheng et al. (2017), covering the time period
from 780 BC to 2015 AD. Additionally, we incorporated earthquakes from the Global
CMT catalog spanning the period from 2016 to 2023 into the dataset. The regressed b-
value was approximately 0.96, with completeness times for magnitudes $M_W$4.5, $M_W$5.0,
$M_W$5.3, $M_W$5.7, $M_W$6.1, and $M_W$6.4 identified as 1985, 1966, 1928, 1916, 1916, and
1900, respectively. It's worth noting that the calculated b-value is slightly higher than
the value of 0.86 reported in Rong et al. (2020), likely due to the inclusion of new
earthquakes occurring after 2015. Figure 5c provides a visualization of the Gutenberg-
Richter relationship in the East Tibet Seismic Zone, in which the b value is 0.96.




### 3.2 **Comparison of Modeled Results**

We depicted the NMS ratios and modeled seismicity rates in Figure 6. The right panels showcased the NMS ratios of the segments in Model 1~4. Model 1 exhibited the most balanced results between the modeled seismicity rates and historical ones. In Figure 6a, all segments in Model 1 demonstrated NMS ratios smaller than 30%. Chartier et al. (2019) suggested that NMS ratios below 30%-40% serve as a benchmark to assess the validity of multi-segment combination models, indicating effective consumption of the slip rate of each segment into seismicity rates for each fault segment. The left panels in Figure 6b further underscored the harmony between the modeled and observed seismicity rates. Here, the observed historical seismicity rates closely aligned with the calculated ones, particularly for <M7 earthquakes.

Compared to Model 1, Model 2 combined segments F17~F20 as a single unit, instead of considering the F17~F18 segments and the F19~F20 segments separately. The left panel of Figure 6b indicates that the NMS ratios for segments of F11~F14 and F4~F5 are all greater than 40%, while the F6~F7 multi-segment combination has an NMS ratio ranging from 30% to 40%, showing that the combination of segments F17~F20 has an impact on the seismicity rates of these faults. From the right panel in Figure 6b, the historical seismicity rates for each fault segment were similar to those in Model 1. However, the calculated seismicity rates for each segment in Model 2 became smaller than those in Model 1, except for a slightly higher rate in the magnitude range of 6.0~6.5. This result indicated that the fault slip rates are not being adequately accounted for in Model 2, unlike in Model 1 (Figure 6a).

In Model 3, the rupture combination comprised segments of F30~F35, rather than considering them separately as F30~F32 and F33~F35. Most segments exhibited high NMS ratios in the left panel of Figure 6c. The calculated seismicity rates were generally





smaller than the historical ones in the right panel of Figure 6c. Similarly, Model 4 was
utilized to investigate whether the great earthquakes of the Y-shaped rupture, combining
segments F4~F10 with F11~F14, could occur. The NMS ratios for each segment and
the calculated seismicity rates were comparable to those observed in Model 3 (Figure
6d).

In addition, we also presented the results using the rupture scaling relationship

proposed by Wells and Coppersmith (1995) in Figure S1. Model 1 exhibited the most
consistent outcomes, with the maximum NMS ratio observed on F14 at 39.3%. The
NMS ratios for all other segments were below 30%. For the calculated seismicity rates
obtained from Model 2 to Model 4 using the rupture scaling relationship of Wells and
Coppersmith (1995), we observed similar patterns. All three models showed segments
with NMS ratios exceeding 40%. Furthermore, we found that these models utilizing the
rupture scaling of Wells and Coppersmith (1995) consistently yielded higher NMS
ratios on average compared to those obtained from the rupture scaling of Cheng et al.

(2020).



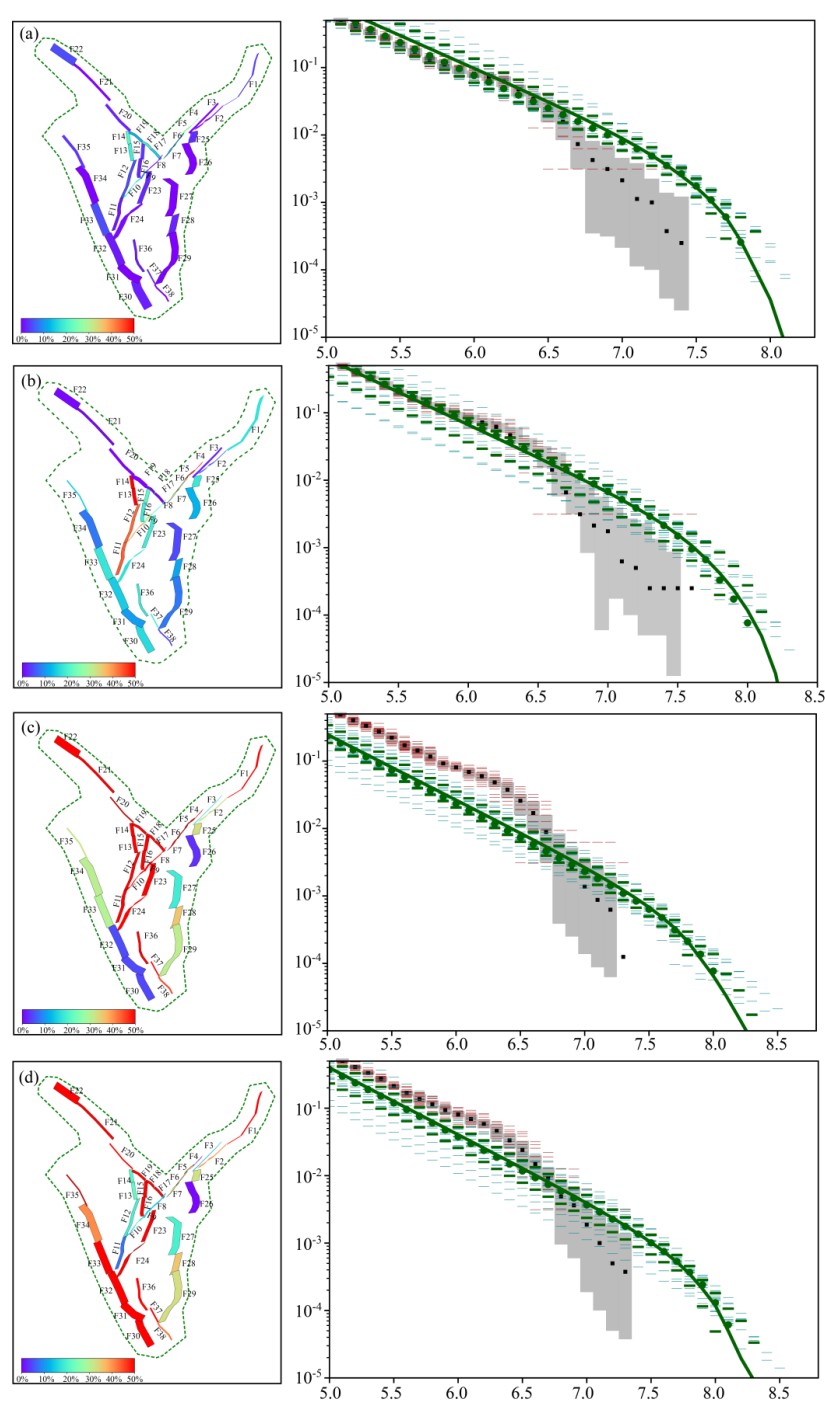

Figure 6 Calculated NMS ratios and comparison results for different models using the



G-R relation. (a) Modeled Non-Mainshock Slip (NMS) Ratio; (b) Comparisons
between the historical Seismicity rates for different models. Dashed green lines are the
MFD of each model, and the solid green line is the mean MFD, green patches represent
the uncertainty (16-84 percentiles). The dotted black line is the rate from the catalog;
the dashed red lines are sampled rates of the catalog exploring the uncertainties on the
magnitudes of earthquakes, and gray rectangular show the one-sigma uncertainty on the
earthquake rates in statistical analysis.
Based on the comparison among different rupture combination models, Model 1
demonstrated the most consistent results among the multi-segment rupture
combinations, fault segment slip rates, and the Magnitude-Frequency relationship.
Therefore, we utilized the seismicity rates from Model 1 to calculate the Peak Ground
Acceleration (PGA) values for the NWYR.

**3.3 Comparison with the results of current national seismic hazard map**
We utilized the OpenQuake Engine v3.10 (Pagani et al., 2014) to calculate the
Peak Ground Acceleration (PGA) values for the NWYR. In this computation, we
employed a logic tree model comprising the Abrahamson et al. (2014); Chiou and
Youngs (2014); Campbell and Bozorgnia (2014); and Boore et al. (2014) branches.
Each branch was assigned an equal weight of 0.25, following the selection criteria
established by Dangkua et al. (2018) for mainland China. These Ground Motion
Prediction Equations (GMPEs) are tailored for earthquakes characterized by moment
magnitude ($M_W$) and the distance to the rupture plane ($R_{rup}$) or its surface projection
($R_{JB}$).
Figure 7a illustrated the distribution of Peak Ground Acceleration (PGA) for the




site condition of firm to hard rock (Vs30=760 m/s, or NEHRP B) resulting from the
seismicity model in Model 1, corresponding to a 10% probability of exceedance in 50
years, which is equivalent to a return period of 475 years. The analysis revealed
concentrations of high values exceeding 0.40 g near fault sources, particularly in areas
with multiple fault sources. These areas include the F2~F5 segments of the Lijiang-
Xiaojinhe fault, the vicinity of the Yulong East Fault (YLE-Fau), the southern part of
the Zhongdian fault (ZD-Fau), and the northern extent of the Heqing-Eryuan fault (HE-
Fau).

The area of the F2~F5 segments includes three parallel faults, with the sum of the

strike-slip of ~3 mm/yr, makes the PGA values relatively higher. The maximum
magnitude of the combinations of F17~F18 segments and the F15~F16 segments are
both approximately $M_W$6.6. These areas exhibit a prevalence of moderate earthquakes
with short recurrence intervals and high Peak Ground Acceleration (PGA) distributions
over a 475-year period. The modeled seismicity rates of the F23 segment and the F24
segment both complied with the G-R relationship, containing enough intermediate
earthquakes, induced the high PGA values around. Along the Chenghai fault, high PGA
values are also observed around the F27~F28 segments with strike-slip rates of 3.0
mm/yr but are lower around the F29 segments with a strike-slip rate of 2.5 mm/yr. For
the Red River fault and its extensions, including the Tongdian-Weishan fault and the
Weixi-Qiaohou fault, high PGA values were concentrated around the F37~F38
segments and the intersection points of the F11, F32, and F32 segments.

Comparison with the national seismic hazard map of the China Seismic Ground

Motion Parameters Zonation Map (CSGMPZM) (Gao et al., 2015) (Figure 7b) for the
site condition of dense soil and soft rock (Vs30 = 500 m/s, or NHERP C) (Chen et al.,
2021), our Peak Ground Acceleration (PGA) values are consistently much higher and



more intricate. The Vs30 of 500 m/s is equivalent to the Type II in the classification
table of CSGMPZM, while the value of 760 m/s belongs to Type I1. Table 1 was the
adjustment factors used by CSGMPZM for site amplification (Gao et al., 2015). Even
if we applied these site amplification adjustment factors to convert our PGA values
from type I1 to type II, the PGA values would not change obviously as the adjustment
are the near to 1 for PGA values of 0.30~0.40 g, and 1 for PGA values of ≥0.40 g. In
figure 7b, the CSGMPZM indicates two high PGA values ranging from 0.30 to 0.40 g
in the NWYR, specifically around the F23~F24 segments and the F27~F28 segments,
respectively. PGA values in other areas surrounding the fault segments in our model
range from 0.20 to 0.30 g. In the development of the CSGMPZM, the region in the
around China was divided into 29 large seismic source zones to calculate the parameters
of the Magnitude-Frequency Distribution (MFD). Additionally, over 1,000 potential
fault sources across China were incorporated into the model. Historical seismicity rates
on the MFD were employed to predict future seismic activity. This methodology led to
lower anticipated seismicity rates in regions with limited historical earthquake records.
The identification of potential fault sources in the CSGMPZM relied on expert opinions
gleaned from research on historical surface rupture, fault segmentation, and the
distribution of past earthquakes. These data sources were subsequently utilized to
allocate predicted seismicity rates based on the MFD. Furthermore, the utilization of
different Ground Motion Prediction Equations (GMPEs) in the CSGMPZM compared
to our results could also contribute to variations in PGA values. The CSGMPZM
utilized GMPEs from Yu et al. (2013) based on surface magnitude ($M_S$) and epicentral
distance ($R$epi). Their GMPEs result in higher PGA values for distances less than 80
km but lower values for distances ≥80 km (Cheng et al., 2021). Consequently, the
seismicity rates derived from the fault slip rates and the multi-segment rupture



Natural Hazards Open Access
and Earth System
Sciences
Discussions



combinations were key factors that rendered our modeled PGA values higher than those
from CSGMPZM.

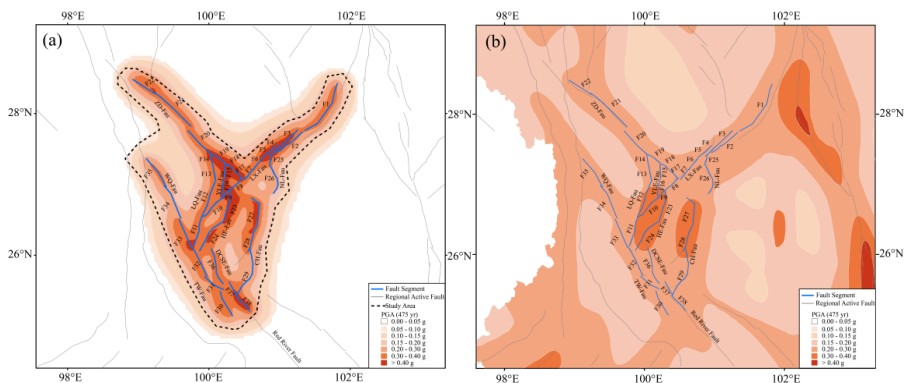

Figure 7 Comparison of the Modeled PGA distribution of 10% in the next 50 years,
(a) the PGA results in this article (b) the PGA results in the CSGMPZM.
Table 1 Adjustment factors for PGA values of different Site condition via Type II

| PGA values | Site condition type | | | | |
|---|---|---|---|---|---|
| for type II | $I_0$ | $I_1$ | II | III | IV |
| ≤0.05 g | 0.72 | 0.80 | 1.00 | 1.30 | 1.25 |
| 0.10 g | 0.74 | 0.82 | 1.00 | 1.25 | 1.20 |
| 0.15 g | 0.75 | 0.83 | 1.00 | 1.15 | 1.10 |
| 0.20 g | 0.76 | 0.85 | 1.00 | 1.00 | 1.00 |
| 0.30 g | 0.85 | 0.95 | 1.00 | 1.00 | 0.95 |
| ≥0.40 g | 0.90 | 1.00 | 1.00 | 1.00 | 0.90 |


In Figure 8, we further illustrated seismicity rates for several typical fault segments
to elucidate the reasons behind the observed high PGA values. In Figure 8a, the
seismicity rates of the F2 segment exhibit a typical G-R relationship, leading to a high
PGA distribution in the surrounding area. We compared the seismicity rates on the
F7~F8 segments, and the F10 segment with the recurrence intervals from paleo-
earthquake studies. In Figures 8b~8c, the red bars illustrate that our modeled seismicity





rates align with the recurrence interval of approximately 3000 years for a magnitude
7.5 earthquake, as determined by Ding et al. (2018). We have also observed that
segments F7 to F8 of the Lijiang-Xiaojinhe fault tend to conform to the characteristic
earthquake model based on their seismicity rate distribution. Segment F10, with a
length of approximately 44 km, experienced rupture during the 1751 M6.8 earthquake.
Tang et al. (2014) identified three paleo-earthquake events with a recurrence interval of
around 5300 years for earthquakes of $M$6.5+ earthquakes on segment F11. They
suggested that the two paleo-events before 1751 AD were considerably stronger than
the one in 1751 AD, implying multi-segment rupturing involving combinations of
segments F11+F12, F11~F13, and F11~F14 resulting in magnitudes of $M_W$7.4~7.6
earthquakes. Additionally, we illustrate the seismicity rates on segments F15, F27, F29,
and F36 in Figures 8e~8f, which closely resemble the G-R distribution, leading to high
PGA distributions in their vicinity. These results demonstrate that the occurrence rate
of intermediate earthquakes influences the high PGA distributions.


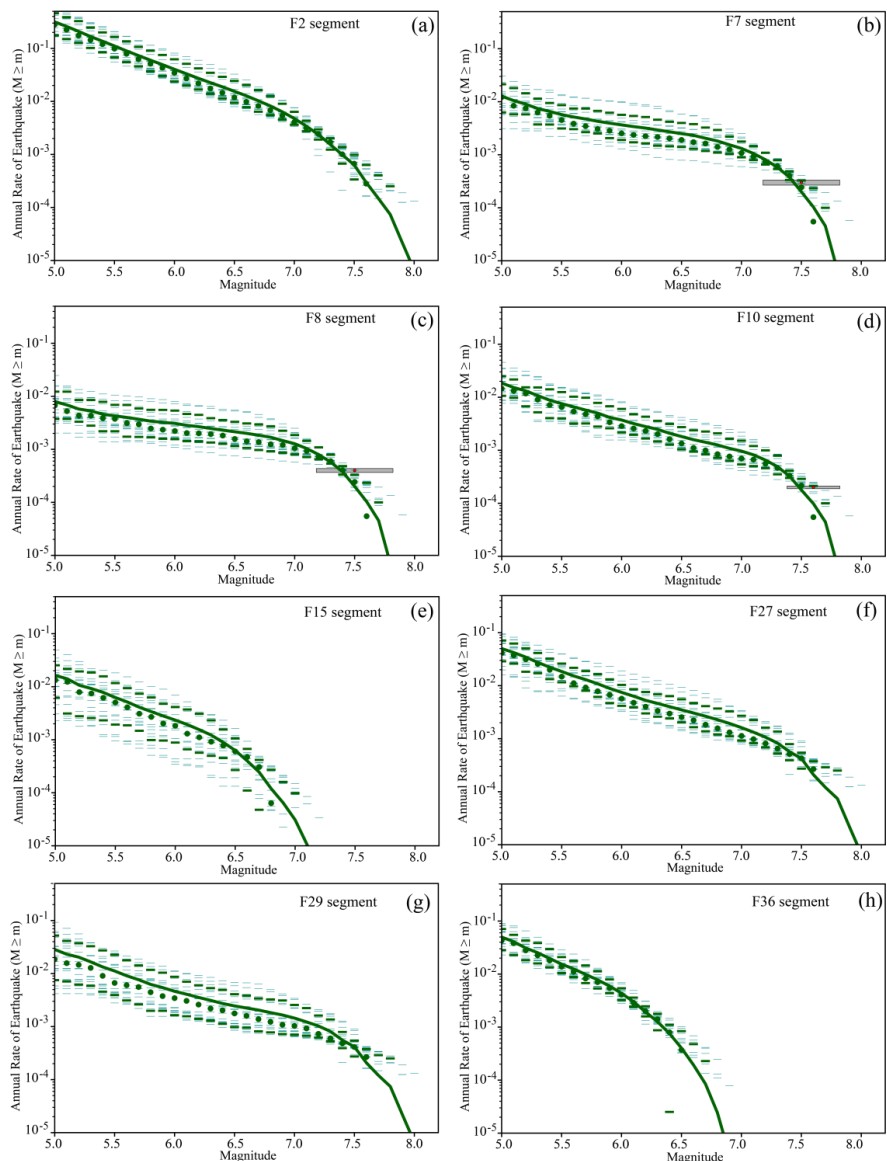

Figure 8 Modeled Seismicity rates for different magnitude on the fault segments. The solid line is the mean MFD, and small patches represent the uncertainty (16-84 percentiles). The dotted line is the rate from the catalog with uncertainties. The red circle is the occurrence rate of the repeated large historical earthquake rate, and the gray box is the associated uncertainty.




### 3.4 Landslide Probabilities

Utilizing the modeled PGA values for rock site conditions presented in Figure 7a
as a foundation, we enhanced our analysis by incorporating the site amplification effect
derived from Chen et al.'s (2021) comprehensive site condition map. Their research,
leveraging geological unit data, culminated in a detailed site condition map covering
mainland China. Leveraging this invaluable resource, we integrated their site condition
map along with the amplification factors for each geological type compared to type II
(referenced in Table 1) to refine the PGA value distribution map (Figure 9a). Our
methodology involved multiplying the PGA values for specific site conditions by the
ratio of type II PGA values to those of the specific type. This approach effectively
magnified PGA values across different site conditions, enriching the granularity of our
analysis. Figure 9a illustrates the resultant PGA distribution map, now encompassing
site amplifications specifically tailored for the NWYR region. Notably, our findings
reveal minimal alterations in the PGA distribution, particularly in proximity to fault
lines, where PGA values remain consistent or exceed 0.4 g (as detailed in Table 1).
Using simulated ground motion data from potential earthquake scenarios, we
conducted a thorough assessment of landslide susceptibility in the affected regions. Our
analysis employed a machine learning framework, following the methodology outlined
by Xu et al. (2019), to develop a predictive model for earthquake-induced landslides.
This model was trained utilizing data from nine earthquakes, ranging from the 1999
$M_W$7.7 Chichi earthquake to the 2017 $M_W$6.5 Jiuzhaigou earthquake, all of which
occurred within or near China. The training dataset comprised samples of earthquake-
induced landslides alongside 13 relevant factors. These factors encompassed diverse
parameters such as elevation, slope angle, slope aspect, land cover, proximity to faults,





geological characteristics, average annual rainfall, and PGA. Leveraging this rich
dataset, we constructed a robust predictive model capable of discerning landslide
probabilities.
Figure 9b illustrates the resultant landslide probability map for the NWYR region.
Notably, areas exhibiting high PGA distribution correspond closely to regions with
elevated landslide probabilities. For instance, notable areas include the northern end of
the Zhongdian fault, the Jinpingshan fault, the Yulong East fault, the northern end of
the Heqing-Eryuan fault, the northern part of the Chenghai fault, and the eastern section
of the Lijiang-Xiaojinhe fault (the F2~F4 segments). Of particular significance are
regions surrounding the Yulong East fault and the convergence zone of the Lijiang-
Xiaojinhe fault and the Zhongdian fault. These areas exhibit pronounced differences in
altitude, ample rainfall, and elevated PGA values, making them particularly susceptible
to landslide occurrences.
By integrating multiple geospatial factors and leveraging advanced machine
learning techniques, our analysis provides valuable insights into landslide susceptibility
in earthquake-prone regions, aiding in effective risk management and mitigation
strategies.
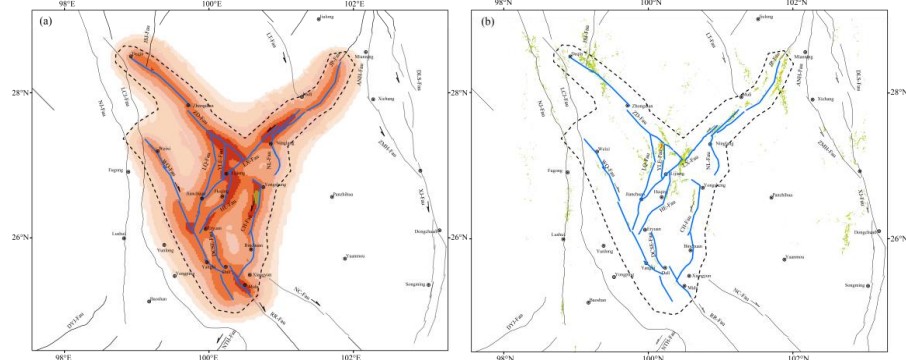
Figure 9. (a) PGA distribution Map considering different site amplifications; (b) the
probabilities of landslide occurrence impacted by the PGA values.


**4 Discussion and Conclusion**

In seismic hazard analysis, understanding fault slip behaviors, including slip rates and fault geometries, is pivotal for accurately modeling future seismicity rates. Concurrently, historical earthquake occurrence rates provide a foundation for estimating these future rates. Of significant note, attention must also be directed towards earthquakes involving multi-segment ruptures, which may not be documented in historical records. In this article, we unveil a new seismic hazard model for the NWYR, where the boundary of the Tibetan Plateau intersects with local low-crustal flow.

**4.1 Multi-segment Rupturing Hazards in NWYR**

The complex fault system results in earthquake occurring almost all the faults with various rupture behaviors in the NWYR, while the catalog of historical and paleo-earthquake data only recorded a small part of the rupturing events. The NWYR serves as the boundary region between China and Myanmar. This area is predominantly inhabited by ethnic minorities in China, resulting in limited written documentation of its history, particularly regarding earthquake disasters. However, some significant earthquakes have been documented, particularly those that have had a seismic impact on major cities like Dali, e.g., the 1515 $M$7.8 earthquake in Yongsheng ruptured two continuous segments of the Chenghai fault (Institute of Geology-State Seismological Bureau and Yunnan Seismological Bureau, 1990). The historical earthquake catalog used in our seismic hazard modeling often struggles to include all these combinations of ruptured scenarios that occurred in the past. What we have done is to search for possible rupture combinations and calculate their seismicity rates to include in our model. These rupture combinations might be constrained by various factors, such as the geometry of fault segments, the width of the step-over between each pair of segments, and the maturity of the fault steps (Cunningham and Mann, 2007; Biasi and Wesnousky,




2017). For strike-slip faults, a width of 5 km is often used to assess the reasonableness
of the rupture combinations (e.g., Biasi and Wesnousky, 2017). However, in the NWYR,
where faults are located in the conduit of ductile low-crust flow, all step-overs have
widths of less than 5 km except the one of ~ 7 km between F20 and F21 segments.
Hence, we advocate that the intersection relationship between faults is the primary
determinant of whether multi-segment rupture events occur among fault segments in
this region.
**4.2 Implication of the small-block rotation in NWYR**
The Holocene strike-slip motion of the Lijiang-Xiaojinhe fault behaves the
dominant role in this region, and intersects the Heqing-Eryuan fault and the Yulong East
fault. Model 1 also confirmed the capability of the entire rupture of the F4~F10
segments of the Lijiang-Xiaojinhe fault. The Chenghai fault and the Zhongdian fault
also are separated by the Lijiang-Xiaojianghe fault, which differs the view of Wang et
al. (1998) that the Dali fault (including the Longpan-Qiaohou fault and the Chenghai
fault) is the primary fault in this region. The Longpan-Qiaohou fault obstructs the
westward continuation of the Lijiang-Xiaojinhe fault, and simultaneously, the F11
segment also resists rupturing in conjunction with the Lijiang-Xiaojinhe fault (Model 4
in Figure 4). In contrast, the Weixi-Qiaohou fault (WQ-F) and the Tongdian-Weishan
fault (TW-F) are part of distinct small-blocks and therefore cannot rupture
simultaneously, as depected in Figure 10a. This indicates that the northern end of the
Red River fault is intercepted by the Longpan-Qiaohou fault. The Zhongdian fault (ZD-
F) was separated to rupture in our model (Model 1 in Figure 4), especially for the
F17~F18 segments combination and the F19~F20 segments combination. Here, we
propose that the normal- and strike-slip of the Yulong East fault poses a greater
destructive potential to the Zhongdian fault compared to the strike-slip of the Longpan-





Qiaohou fault.

Hence, our configurations of multi-segment ruptures portrayed in Model 1 of

Figure 4 correspond to the rotational patterns noted in the small block delineated in the
NWYR by Wang et al. (1998). We illustrated this clockwise rotation of the small
blocks in the NWYR in Figure 10a. This clockwise rotation was further supported by
GPS observations to the west of the Xianshuihe fault, the Anninghe fault, and the
Xiaojiang fault, after eliminating the entire movement (Figure 10b) (Cheng et al., 2012).
In Figure 10b, the area where the Nujiang fault intersects with the Dayingjiang fault
experiences the strongest extensional forces. Rangin et al. (2013) and Lindsey et al.
(2023) proposed that the dynamic source of this extensional tectonic environment was
the side effect of the gravitational collapse of the Tibetan Plateau with the westwards
of upper crust faster than the lower crust (Rangin et al., 2013; Lindsey et al., 2023).
This extensional force exerts obviously on the faults in our model, making the rotation
of small blocks, and the normal slip of the regional faults, e.g., the Diancangshan fault
and the Chenghai fault.

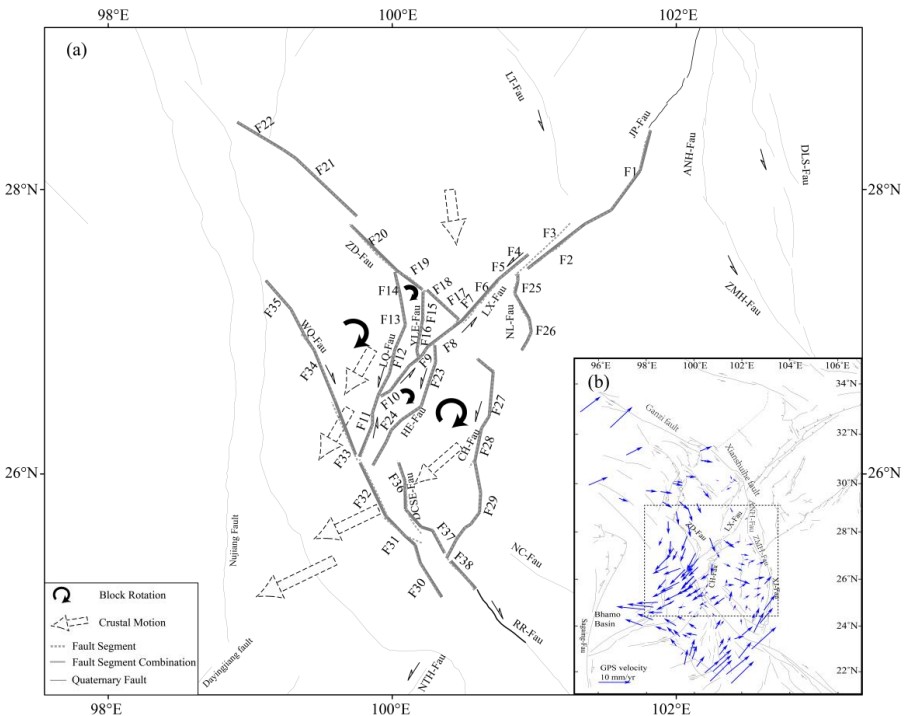

Figure 10 a. Kinematic Model of the faults in the NWYR; b. Regional GPS

motion after removing the whole movement (Cheng et al., 2012).

In conclusion, our study has provided valuable insights into the seismic hazard

present in the NWYR. By developing fault segmentation models based on recent

geological research and utilizing advanced simulation techniques, we have enhanced

our understanding of fault activities and seismicity rates across the region. Through

careful analysis and consideration of fault segmentation, fault slip rates, and historical

seismicity, we have identified multi-segment models that best represent the observed

data. Our calculations of PGA with a 10% probability of exceedance within 50 years

offer crucial information for assessing seismic risk in the NWYR. The PGA values,

associated with obvious latitude difference, abundant precipitation, are prone to

occurrence of landslides.



Furthermore, our investigation into multi-segment rupture combinations has
illuminated potential scenarios for seismic events in the region. Through the integration
of these findings, we have generated a more comprehensive assessment of seismic
hazards and landslide probabilities. These are intertwined with the regional small block
rotation induced by the low-crustal flow and gravitational collapse along the
southeastern frontier of the Tibetan Plateau.
Moving forward, continued research and monitoring efforts are essential for
refining our understanding of seismic hazards in the region. Further investigations into
fault behaviors, fault interactions, and the potential for multi-segment ruptures will be
crucial for enhancing the accuracy and reliability of seismic and induced geological
hazard assessments. By remaining vigilant and proactive in our approach to seismic risk
management, we can better protect communities and infrastructure in the face of future
seismic events in the NWYR and beyond.

*Code availability*
In this study, we have used the code related to Chartier et al. (2019,
https://doi.org/10.1785/02201803320), which can be downloaded from the webpage
(https://doi.org/10.1785/02201803320, last accessed in May, 2024).

*Data availability*
The focal mechanism data are from Global CMT catalog (www.globalcmt.org, last
accessed in May, 2024) Table S1 in the supplementary material for this paper includes
the fault segments, historical and paleo-earthquakes and their associated slip parameters.



*Author contributions.*

Jia Cheng was responsible for methodology, software, and writing the original draft.

Chong Xu worked for the landslide occurrence probabilities calculation. Xiwei Xu

and Shimin Zhang contributed to design the fault rupture combination models.

Pengyu Zhu contributed to seismic hazard modeling.

*Competing interests.*

The authors declare that they have no known competing financial interests or personal

relationships that could have appeared to influence the work reported in this paper.

*Acknowledgments.*

We thank Dr. Guangwei Zhang from National Institute of Natural Hazards and Dr.

Mingming Jiang from Institue of Geology and Geophysics, Chinese Academy of

Sciences for discussion on the dynamic source of the crustal deformation. We are also

grateful to Mr. Rui Ding from National Institute of Natural Hazards and Dr. Panxing

Yang from Institute of Earthquake Forecasting, China Earthquake Administration for

their assistance in delineating fault traces and the fault segmentation work.

*Financial support.*

This study receives funds from the National Natural Science Foundation of China (No.

U2039201 and No. 42074064), and National Institute of Natural Hazards, Ministry of

Emergency Management of China (Grant NO. ZDJ2020-14).

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
