# Peer review of "Modeling Seismic Hazard and Landslide Potentials in Northwestern Yunnan, China: Exploring Complex Fault Systems with multi-segment rupturing in a Block Rotational Tectonic Zone"

_Natural Hazards and Earth System Sciences, 2024_

## Referee Comment (RC1)

**Referee Report** on "*Modeling Seismic Hazard and Landslide Potentials in Northwestern Yunnan, China: Exploring Complex Fault Systems with Multi-segment Rupturing in a Block Rotational Tectonic Zone*"

**Summary**:

The paper entitled "*Modeling Seismic Hazard and Landslide Potentials in Northwestern Yunnan, China: Exploring Complex Fault Systems with Multi-segment Rupturing in a Block Rotational Tectonic Zone*" is authored by Jia Cheng, Chong Xu, Xiwei Xu, Shimin Zhang, and Pengyu Zhu. The affiliations are the School of Earth Science and Resources, China University of Geosciences (Beijing), and the National Institute of Natural Hazards, Ministry of Emergency Management of China, both located in Beijing, China. The corresponding author is Jia Cheng.

**Strengths and Novelties**:

The study addresses a significant gap in understanding the seismic hazards in the Northwestern Yunnan Region (NWYR) by focusing on the complex fault systems and their potential for multi-segment rupturing. The integration of fault slip parameters and the assessment of multi-segment rupturing risks using four potential models is particularly innovative. The analysis identifies Model 1, which focuses on multi-segment rupture combinations on single faults, especially the Zhongdian fault, as the most suitable for the NWYR. This model is validated by the alignment of modelled seismicity rates with fault slip rates. The use of peak ground-motion acceleration values, calculated with a 475-year return period, and their correlation with fault distribution provides a detailed understanding of the seismic hazard landscape. Furthermore, the study's simulation of landslide occurrence probabilities, using peak ground-motion acceleration distribution maps, highlights the intricate interplay between multi-segment rupturing hazards and regional geological dynamics. This integration of seismic hazard modelling with landslide probabilities is a notable strength of the paper.

**Weaknesses and Areas for Improvement**:

While the study is comprehensive, there are several areas that require improvement to enhance the manuscript's quality. Firstly, the introduction could benefit from a more detailed literature review to contextualise the current study within the broader field of

seismic hazard analysis. Secondly, the methodology section, although detailed, could be more clearly structured to ensure readers can easily follow the complex modelling processes. Additionally, while the study highlights the importance of fault segmentation and multi-segment rupturing, it would benefit from a more explicit discussion of the limitations of the models used and the assumptions made during the simulations.

**Constructive Criticism and Suggestions for Improvement**:

To improve the manuscript, the authors should consider incorporating the following suggestions:

1. Expand the literature review to include more recent studies on seismic hazard analysis and multi-segment rupturing to provide a comprehensive background for the research.
2. Clarify the methodology section by breaking down the modelling process into more distinct sub-sections, each with clear headings and explanations.
3. Discuss the limitations of the study in greater detail, particularly the assumptions made during the modelling and their potential impact on the results.
4. Include a section on future research directions, highlighting how the current study could be expanded or refined with additional data or more advanced modelling techniques.

**Research Gaps**:

The paper identifies the lack of comprehensive seismic hazard models that integrate fault geometry and segmentation with historical seismicity rates as a significant research gap. While the study makes a substantial contribution towards filling this gap, further research is needed to validate the models used and to explore the potential for other fault systems to exhibit similar multi-segment rupturing behaviour. Additionally, the impact of climate change on landslide probabilities and seismic hazards in the region could be an important area for future investigation.

**Missing References**:

Several relevant references are missing from the current manuscript. These include recent studies on seismic hazard analysis, fault segmentation, and multi-segment rupturing. Incorporating these references would provide a more comprehensive context for the research and strengthen the validity of the study's findings.

Furthermore, I would like to kindly suggest that the authors incorporate references to a few previous studies that seem to have been overlooked. For instance, the phenomenon of multiple ruptures has been applied to the problem of tsunami generation, as demonstrated in the following article: Dutykh, D., Mitsotakis, D., Gardeil, X., & Dias, F. (2013). On the use of the finite fault solution for tsunami generation problems. *Theoretical and Computational Fluid Dynamics*, *27*(1–2), 177–199. https://doi.org/10.1007/s00162-011-0252-8. Additionally, probabilistic methods have been applied to tsunami hazard assessment, as illustrated in the manuscript: Rashidi, A., Shomali, Z. H., Dutykh, D., & Keshavarz Farajkhah, N. (2020). Tsunami hazard assessment in the Makran subduction zone. *Natural Hazards*, *100*(2), 861–875. https://doi.org/10.1007/s11069-019-03848-1. It would be beneficial for the authors to examine the approaches utilised in the tsunami wave community and compare them with the methodologies applied in their study of landslide hazards. Incorporating these references will not only strengthen the context of the research but also provide a broader perspective on multi-segment rupture phenomena and probabilistic hazard assessment.

**Language and Grammar Corrections**:

The manuscript contains several language and grammar errors that need correction. Here are some identified issues:

1. Page 3, Line 45: "the Eurasia Platea" should be "the Eurasian Plate."
2. Page 3, Line 46: "Plateau world highest" should be "Plateau, the world's highest."
3. Page 5, Line 80: "diverse rupture behaviors contributes" should be "diverse rupture behaviors contribute."
4. Page 6, Line 108: "resulting in notable errors" should be "resulting in significant errors."
5. Page 8, Line 160: "increased precision and reliability" should be "increasing precision and reliability."

**Conclusion**:

In conclusion, the paper presents a valuable contribution to the understanding of seismic hazards and landslide potentials in the Northwestern Yunnan Region. However, several areas need improvement, particularly in terms of literature review, methodology clarity, and addressing limitations. By incorporating the suggested revisions and adding the missing references, the authors can significantly enhance the manuscript's quality. I recommend a revision of the paper to address these points.

This report is intended to provide constructive feedback to the authors to help them improve their work and to ensure that the manuscript meets the high standards required for publication in the Natural Hazards and Earth System Sciences (NHESS) Journal.

---

## Author Comment (AC1)

Reviewers' comments:

**Referee 1**

**1. Weaknesses and Areas for Improvement:**

While the study is comprehensive, there are several areas that require improvement to enhance the manuscript's quality. Firstly, the introduction could benefit from a more detailed literature review to contextualise the current study within the broader field of seismic hazard analysis. Secondly, the methodology section, although detailed, could be more clearly structured to ensure readers can easily follow the complex modelling processes. Additionally, while the study highlights the importance of fault segmentation and multi-segment rupturing, it would benefit from a more explicit discussion of the limitations of the models used and the assumptions made during the simulations.

Thanks for your review.

1. Expand the literature review to include more recent studies on seismic hazard analysis and multi-segment rupturing to provide a comprehensive background for the research.

Thanks for your recommendation! We added the recent studies on multi-segment rupturing seismic hazard analysis in Section 3. See Line 339 in our modified version.

2. Clarify the methodology section by breaking down the modelling process into more distinct sub-sections, each with clear headings and explanations.

Thanks for your suggestion!    We divided the methodology part into section 3.1 and section 3.2.

3. Discuss the limitations of the study in greater detail, particularly the assumptions made during the modelling and their potential impact on the results.

Thanks! We added section 4.1 of "Model limitations and mitigation measures".

4. Include a section on future research directions, highlighting how the current study could be expanded or refined with additional data or more advanced modelling

techniques.

Thanks for your suggestion!    We added the future research directions in the last paragraph in Line 737-746.

**2.  Research Gaps:**

The paper identifies the lack of comprehensive seismic hazard models that integrate fault geometry and segmentation with historical seismicity rates as a significant research gap. While the study makes a substantial contribution towards filling this gap, further research is needed to validate the models used and to explore the potential for other fault systems to exhibit similar multi-segment rupturing behaviour. Additionally, the impact of climate change on landslide probabilities and seismic hazards in the region could be an important area for future investigation.

Thanks for your suggestion! We will focus on the impact of climate change on landslide probabilities and seismic hazards in the region in the next studies.

**3.  Missing References:**

Several relevant references are missing from the current manuscript. These include recent studies on seismic hazard analysis, fault segmentation, and multi-segment rupturing. Incorporating these references would provide a more comprehensive context for the research and strengthen the validity of the study's findings.

Furthermore, I would like to kindly suggest that the authors incorporate references to a few previous studies that seem to have been overlooked. For instance, the phenomenon of multiple ruptures has been applied to the problem of tsunami generation, as demonstrated in the following article:

Dutykh, D., Mitsotakis, D., Gardeil, X., & Dias, F. (2013). On the use of the finite fault solution for tsunami generation problems. Theoretical and Computational Fluid Dynamics, 27(1−2), 177−199. https://doi.org/10.1007/s00162-011-0252-8.

Additionally, probabilistic methods have been applied to tsunami hazard assessment,

as illustrated in the manuscript: Rashidi,

A., Shomali, Z. H., Dutykh, D., & Keshavarz Farajkhah, N. (2020). Tsunami hazard assessment in the Makran subduction zone. Natural Hazards, 100(2), 861−875. https://doi.org/10.1007/s11069-019-03848-1.

It would be beneficial for the authors to examine the approaches utilised in the tsunami wave community and compare them with the methodologies applied in their study of landslide hazards. Incorporating these references will not only strengthen the context of the research but also provide a broader perspective on multi-segment rupture phenomena and probabilistic hazard assessment.

Thanks! We added these studies as the reference work in Line 332 to Line 354. We also referred the works of multi-segment rupturing on tsunamic studies in Line 348 and Line 349.

**4. Language and Grammar Corrections:**

The manuscript contains several language and grammar errors that need correction.

Here are some identified issues:

1. Page 3, Line 45: "the Eurasia Platea" should be "the Eurasian Plate."

Modified in Line 45.

2. Page 3, Line 46: "Plateau world highest" should be "Plateau, the world's highest."

Thanks! We modified it in Line 45.

3. Page 5, Line 80: "diverse rupture behaviors contributes" should be "diverse rupture behaviors contribute."

Modified in Line 80.

4. Page 6, Line 108: "resulting in notable errors" should be "resulting in significant errors."

Thanks! We modified it. See Line 107.

5.  Page 8, Line 160: "increased precision and reliability" should be "increasing precision and reliability."

Modified in Line 156.

---

## Author Comment (AC2)

Reviewers' comments:

**Referee 1**

**1.  Weaknesses and Areas for Improvement:**

While the study is comprehensive, there are several areas that require improvement to enhance the manuscript's quality. Firstly, the introduction could benefit from a more detailed literature review to contextualise the current study within the broader field of seismic hazard analysis. Secondly, the methodology section, although detailed, could be more clearly structured to ensure readers can easily follow the complex modelling processes. Additionally, while the study highlights the importance of fault segmentation and multi-segment rupturing, it would benefit from a more explicit discussion of the limitations of the models used and the assumptions made during the simulations.

Thanks for your review.

1.  Expand the literature review to include more recent studies on seismic hazard analysis and multi-segment rupturing to provide a comprehensive background for the research.

Thanks for your recommendation! We added the recent studies on multi-segment rupturing seismic hazard analysis in Section 3. See Line 339 in our modified version.

"Numerous studies have focused on understanding the fault's geometric and physical parameters to ascertain conditions conducive to multi-segment rupturing. Factors identified include step width (e.g., < 5 km) (Harris and Day, 1999; Lozos et al., 2012), fault structural maturity characterized by initiation age, net slip, length, and slip rate (Manighetti et al., 2007; 2021), and geometric irregularities such as fault branches and bends, significantly influenced by the pre-existing stress field (Mignan et al., 2015). Recognizing the significance of these rupture parameters in producing multi-segment

rupturing, recent studies, such as those by Chatier et al. (2019), Cheng et al. (2021), Lee et al. (2022), and Chang et al. (2023), included the possibilities and probabilities of multi-segment rupturing in seismic hazard analysis. Additionally, Dutykh et al. (2013) and Rashidi et al. (2020) employed multi-segment rupturing into models of tsunami wave generation. The concept of multi-segment rupturing was also incorporated in the UCERF3 model through their complex "Grand Inversion" methodology, which integrates data on fault slip rates, historical seismicity, and paleoseismic records (Page et al., 2014). However, for most other regional studies, collecting all the necessary input parameters remains challenging. "

2. Clarify the methodology section by breaking down the modelling process into more distinct sub-sections, each with clear headings and explanations.

Thanks for your suggestion! We divided the methodology part into section 3.1 and section 3.2.

**"3.1 Methodology"** and **"3.2 Scaling Relationship and Modeling Parameters"**

3. Discuss the limitations of the study in greater detail, particularly the assumptions made during the modelling and their potential impact on the results.

Thanks! We added section 4.1 of "Model limitations and mitigation measures".

"Our seismic hazard modeling for NWYR represents our current understanding of average earthquake hazards in the region based on available data. The results are affected by numerous epistemic and aleatory uncertainties inherent in seismic hazard modeling processes, including the MFD, fault geometry, fault type, slip rate, and variability in GMPEs. Mitigating the impact of these uncertainties is critical for

accurate seismic hazard assessment.

The MFD relationship, calculated from historical earthquakes, is essential for determining seismicity rate ratios across different magnitude bins. The deflection of the MFD directly influences the distribution of the modeled seismicity rates. In this study, we chose the G-R relationship over the Y-C relationship due to the regional fragmented tectonic environment. The calculated $b$-value of 0.96 aligns closely with the expected value of 1 found in seismically active regions (Pacheco et al., 1992). To derive earthquake magnitudes on fault segments, we employed rupture scaling relationships based on historical rupture parameters of earthquakes in China as proposed by Cheng et al. (2020), ensuring consistency with unique tectonic characteristics. Achieving more precise MFDs and rupture scaling laws necessitates further refinement in methodology and the use of reliable catalogs specific to the study area.

For fault geometry, type, and slip rates, we relied exclusively on recent field investigation data. In compiling fault rupture models for NWYR, we analyzed these geological data under a unified tectonic stress field, ensuring coordinated fault system movements. The variability in GMPEs is complex, influenced by factors such as earthquake rupture characteristics, seismic wave propagation, and site conditions. Consequently, we incorporated Quaternary sediment site amplification effects on PGA values. Addressing basin effects on ground motion requires dynamic simulations to achieve more precise results."

4. Include a section on future research directions, highlighting how the current study could be expanded or refined with additional data or more advanced modelling

techniques.

Thanks for your suggestion! We added the future research directions in the last paragraph in Line 737-746.

"Future seismic hazard work can be improved by utilizing geophysical data to understand fault structures where strong earthquakes are developing (Xu et al., 2017), applying geodetic data to assess energy accumulation on fault segments (e.g., Yao and Yang, 2023), using microseismicity relocation data to reveal fault asperities (Lay and Nishenko, 2022), and employing dynamic rupture simulations of single and multi-segments to enhance earthquake motion predictions (e.g., Zhang et al., 2017). These studies on fault behaviors, interactions, and multi-segment ruptures are vital for improving seismic hazard assessments. Staying vigilant and proactive in seismic risk management will better protect communities and infrastructure in the NWYR and beyond."

**2. Research Gaps:**

The paper identifies the lack of comprehensive seismic hazard models that integrate fault geometry and segmentation with historical seismicity rates as a significant research gap. While the study makes a substantial contribution towards filling this gap, further research is needed to validate the models used and to explore the potential for other fault systems to exhibit similar multi-segment rupturing behaviour. Additionally, the impact of climate change on landslide probabilities and seismic hazards in the region could be an important area for future investigation.

Thanks for your suggestion! We will focus on the impact of climate change on landslide probabilities and seismic hazards in the region in the next studies.

**3. Missing References:**

Several relevant references are missing from the current manuscript. These include recent studies on seismic hazard analysis, fault segmentation, and multi-segment rupturing. Incorporating these references would provide a more comprehensive context for the research and strengthen the validity of the study's findings.

Furthermore, I would like to kindly suggest that the authors incorporate references to a few previous studies that seem to have been overlooked. For instance, the phenomenon of multiple ruptures has been applied to the problem of tsunami generation, as demonstrated in the following article:

Dutykh, D., Mitsotakis, D., Gardeil, X., & Dias, F. (2013). On the use of the finite fault solution for tsunami generation problems. Theoretical and Computational Fluid Dynamics, 27(1−2), 177−199. https://doi.org/10.1007/s00162-011-0252-8. Additionally, probabilistic methods have been applied to tsunami hazard assessment, as illustrated in the manuscript: Rashidi,

A., Shomali, Z. H., Dutykh, D., & Keshavarz Farajkhah, N. (2020). Tsunami hazard assessment in the Makran subduction zone. Natural Hazards, 100(2), 861−875. https://doi.org/10.1007/s11069-019-03848-1.

It would be beneficial for the authors to examine the approaches utilised in the tsunami wave community and compare them with the methodologies applied in their study of landslide hazards. Incorporating these references will not only strengthen the context of the research but also provide a broader perspective on multi-segment rupture phenomena and probabilistic hazard assessment.

Thanks! We added these studies as the reference work in Line 332 to Line 354. We also referred the works of multi-segment rupturing on tsunamic studies in Line 348 and Line 349.

**4. Language and Grammar Corrections:**

The manuscript contains several language and grammar errors that need correction.

Here are some identified issues:

1. Page 3, Line 45: "the Eurasia Platea" should be "the Eurasian Plate."

Modified in Line 45.

2. Page 3, Line 46: "Plateau world highest" should be "Plateau, the world's highest."

Thanks! We modified it in Line 45.

3. Page 5, Line 80: "diverse rupture behaviors contributes" should be "diverse rupture behaviors contribute."

Modified in Line 80.

4. Page 6, Line 108: "resulting in notable errors" should be "resulting in significant errors."

Thanks! We modified it. See Line 107.

5. Page 8, Line 160: "increased precision and reliability" should be "increasing precision and reliability."

Modified in Line 156.

---

## Author Comment (AC3)

Review of manuscript nhess-2024-96

**General comments**

In my opinion, the manuscript could be accepted after a major revision. It does not make clear what is the problem it tries to solve, and it lacks details about key elements of the methodology (i.e., the use of machine learning to calculate landslide hazard). Moreover, the discussion of the results –and essentially the manuscript itself– focuses on the seismic hazard model, while the title suggests that it is about landslide hazard too. Moreover, the documentation calculation of the landslide hazard should be improved. Moreover, given that landslide hazard modelling and the results with respect to landslide hazard are given so little coverage in the manuscript, please consider revising the title, removing from the manuscript whatever concerns landslide hazard, and focusing on seismic hazard.

Although I am neither an English native speaker nor an English language professional, I believe I have found more than a few instances, where the writing should be improved. Therefore, the manuscript does not meet editorial standards, in my opinion. Please consider having the manuscript edited by an English language professional.

As far as the figures are concerned, which have been published elsewhere and are included in the manuscript as they are or after some modification, please make sure that the reproduction rights have been secured, and inform the editor, or please consider removing them.

**Abstract**

Please considering stating clearly what is the main topic of the paper. It is not clear what is the problem that this paper tries to solve. It states that it presents a new probabilistic seismic hazard assessment model that accounts for multi-segment faults, and that it uses this new model to do landslide hazard assessment. As suggested in line 28-29, the new seismic hazard model makes better predictions of some ground motion intensity measures, which may lead to a better assessment of landslide hazard. Moreover, please consider finishing the abstract with a statement about the implications of the findings.

- Line 1, "Potentials": please consider replacing with "hazard".

Thanks! We revise it to Occurrence Probabilities.

- Lines 21-23: Please clarify why the abstract mentions this historical earthquake.

This earthquake is a typical multi-segment rupturing earthquake, as stated by many studies, such as Huang et al., 2021. We revised the words in Line 23: as demonstrated by the historical multi-segment ruptured 1515 M7.8 Yongsheng Earthquake.

- Line 24, "presented": incorrect tense. Please replace with "presents".

Revised it. Thanks!

- Line 24, "a novel seismic hazard modeling study": please replace with to "a new probabilistic seismic hazard model"

Revised it. Thanks!

- Line 25, "integrating fault slip parameters and assessing multi-segment rupturing risks": Please explain why is this being done by this paper. What is the necessity? A classical PSHA would not do?

Thanks for your question!    We mean that in this region of NWYR, the rugged terrain makes it difficult to find the fault surface tracks. The climate is humid with abundant rainfall, leading to high vegetation cover, severe weathering, and significant damage to fault surface traces.

We modified the words from Line 24 to Line 28 as follows:    This article presents a new seismic hazard modeling study for the NWYR, with recent fault geometrical and slip rates studies, incorporating recent findings on fault geometry and slip rates, and integrating fault slip parameters and historical seismicity rates to assess multi-segment rupturing risks.

- Line 28-29, "emerges as": Please consider replacing with "is proposed as".

Revised it. Thanks!

- Line 28-29, "most suitable": Please explain by which criteria and for which use.

We revised the words:"Among the four potential multi-segment rupture combination models examined, Model 1, characterized by multi-segment rupture combinations on single faults, particularly fracturing the Zhongdian fault, is proposed as the most suitable for the NWYR, given that the normalized misfit scores (NMS) are all below

the 30%~40% threshold, supported by the alignment of modeled seismicity rates with fault slip rates."

- Line 28-29, "supported by the alignment": Please consider replacing with: "as suggested by the agreement".

Thanks for your advice. We modified it to: Model 1, characterized by multi-segment rupture combinations on single faults, particularly fracturing the Zhongdian fault, is proposed as the most suitable for the NWYR, given that the normalized misfit scores (NMS) are all below the 30%~40% threshold, supported by the alignment of modeled seismicity rates with fault slip rates.

- Line 30, "demonstrated": incorrect tense. Please replace with "demonstrates".

Revised it. Thanks!

- Line 30, "peak ground-motion acceleration (PGA) values, calculated with a 475-year

  return period from modeled seismicity rates, exhibited": incorrect terminology. Please consider replacing with: "the peak ground acceleration for a mean return period of 475 years, which is calculated with the developed probabilistic seismic hazard model, has"

Thanks! We revised it.

- Line 32, "fault distribution": Please clarify if the manuscript refers to the spatial distribution of the faults.

- "than the China Seismic Ground Motion Parameters Zonation Map": please consider revising replacing with "than the PGA given by the Chinese seismic ground motion parameters zonation map."

We revised them according to the reviewer's comments. Thanks!

- Line 33: Please give a one-sentence description of the simulations.

Thanks! We revised it as follows:

Furthermore, we utilized PGA values with the Bayesian Probability Method and the Machine Learning Model to predict landslide occurrence probabilities, based on our peak ground motion acceleration distribution map.

- Line 34, "across": Please consider replacing with "as a function of".

Revised it. Thanks!

- Line 37, "highlighted": incorrect tense. Please replace with "highlights".

Revised it.

1. Introduction

Please justify why this study only uses the peak ground acceleration. Please state what are the ground motion intensity measures used in the literature for landslide hazard and for vulnerability to landslides.

Both peak ground acceleration (PGA) values and intensity measures can be utilized for assessing landslide hazards and vulnerabilities, as they indicate the magnitude of the seismic forces on the rock generated by earthquakes. In contrast, other parameters, such as peak ground velocity (PGV), primarily convey velocity information. PGA values can be calculated directly from probabilistic seismic hazard assessment (PSHA) studies, whereas intensity maps require further transformation from PGA values.

Line 156: The use of a machine learning model is suddenly mentioned here. Please consider mentioning it in the title, and in the abstract. Please justify the use of machine learning and consider adding comparisons of this calculation using machine learning with classical methods or cite a reference that validated this method.

Thank you for your advice. We have incorporated it into the abstract. However, we did not include it in the title since it represents only a small part of our results. Instead, we revised the title to: 'Modeling Seismic Hazard and Landslide Occurrence Probabilities in Northwestern Yunnan, China: Exploring Complex Fault Systems with Multi-Segment Rupturing in a Block Rotational Tectonic Zone,' which also implies the simulation work related to landslide occurrence.

Line 158-159: "disaster preparedness… in the area". Indeed, this study may help in this direction, but please consider mentioning in the abstract and in the opening of the introduction that this is also part of the context of this study.

Thanks! We add it in the end of the abstract.

3. Multi-segment rupture hazard Modeling

Line 335: Please consider describing what is the state of the art in probabilistic seismic hazard assessment, and then explain why accounting for the slip rate would be an improvement.

Thank you for your advice. We added the words in the beginning of the section 3 of multi-segment rupture hazard Modeling. We emphasize the importance of fault slip rate rather the historical seismicity rate.

"Recognizing the significance of these rupture parameters in producing multi-segment rupturing, recent studies, such as those by Chatier et al. (2019), Cheng et al. (2021), Lee et al. (2022), and Chang et al. (2023), included the possibilities and probabilities of multi-segment rupturing in seismic hazard analysis. Additionally, Dutykh et al. (2013) and Rashidi et al. (2020) employed multi-segment rupturing into models of tsunami wave generation. The concept of multi-segment rupturing was also incorporated in the UCERF3 model through their complex "Grand Inversion" methodology, which integrates data on fault slip rates, historical seismicity, and paleoseismic records (Page et al., 2014; Field et al., 2014). However, for most other regional studies, collecting all the necessary input parameters remains challenging.

In seismic hazard modeling, fault slip rates can be used instead of historical seismicity data to simulate seismicity rates on faults, as slip rates span multiple seismic cycles of large-magnitude earthquakes and provide estimates of the average earthquake recurrence interval (Youngs and Coppersmith, 1985). We utilize the methodology developed by Chatier et al. (2019) to translate these fault slip rates into seismicity rates, considering both multi-segment and single-segment ruptures."

It is not uncommon to take into account the characteristic earthquake in seismic hazard models. It is not clear why this paragraph mentions this in its opening.

We appreciate the reviewer's comment regarding the mention of characteristic earthquakes in our paragraph. The Y-C model is primarily derived from the characteristic earthquake model, which provides a foundation for understanding seismic hazards. In our study, the SHERIFS code offers two options for magnitude-frequency relationships, and we specifically chose the Gutenberg-Richter (G-R) model due to its robustness and widespread acceptance in the literature.

See Line 379-383. "Therefore, in our analysis of seismicity rates for the whole seismicity rates on the regional faults, we opted to utilize the G-R relation (Gutenberg and Richter, 1944) as the Magnitude-Frequency relationship, rather than the Youngs-Coppersmith (Y-C) relation (Youngs and Coppersmith, 1985)."

Please explain why the manuscript focuses on the estimation of the PGA with a 10% probability of exceedance in 50 years. Please consider discussing the PGA for other annual probabilities of exceedance, and other intensity measures. If the national hazard map is only in terms of PGA for 475 years, please consider comparing the other intensity measures with other hazard models.

As stated in Line 107, 'Due to the high altitude, dense vegetation, and easily weathered conditions, obtaining accurate fault slip rates poses a significant challenge, often leading to considerable uncertainties.' In this region, studies on seismic hazard models are sparse, which is why we compared our results with the national hazard map in terms of PGA for a 475-year return period.

Line 581: The reader may have questions about this method, but its description is missing. The machine learning model is trained using scenarios which include the PGA as an entry parameter. However the footprint of the PGA in a scenario is different from a map of the PGA for a specific return period. Moreover, please state if the landslide hazard calculation accounts for all (or a very wide range) of annual

probabilities of exceedance of the PGA (or for a very wide range of return periods), and not just the PGA for 475 years. If it does not, please explain why.

Thank you for your question. We have added text to clarify this in Section 3.4. The clarification is as follows:

Our model directly assessed the absolute probability of landslide occurrence, represented as the percentage of the landslide area within a region relative to the total area of the region (Shao et al., 2020). As a result, our hazard estimates have a true probabilistic meaning, reflecting the actual probability of landslide occurrence rather than being merely a formal expression of probability. We then calculated the probabilistic seismic susceptibility for a specific point in time within the study area, which produced a probabilistic PGA distribution map. By using this probabilistic PGA map as input for our model, we can estimate the corresponding probability of earthquake-triggered landslide occurrence. We employed these steps as the basis of our approach to calculating the probability of such landslides.

**Conclusions**

Line 671: Please add section title for the conclusions.

Thanks! We added it.

Line 671: The opening statement claims that the manuscript has given insights. This sentence seems out of place, because the manuscript first needs to briefly state the insights, then explain their importance, and then claim that it made valuable insights. Please consider dedicating the biggest part of the conclusions to the importance of the findings.

Thanks for your suggestion. We modified it as follows:

This study presents a comprehensive seismic hazard model for the NWYR, integrating fault slip rates and historical seismicity data to assess the risks of multi-segment ruptures and landslide occurrences. By leveraging fault slip rates and fault geometrical distributions in the NWYR, we employed the iterative method within the SHERIFS code to simulate seismicity rates for both single-segment and multi-segment ruptures. This work underscores the complexity of the fault systems within the region's

block rotational tectonic environment. Our study has yielded valuable insights into the seismic hazards present in the NWYR. Through the development of fault segmentation models based on recent geological research and the application of advanced simulation techniques, we have significantly enhanced our understanding of fault activity and seismicity rates across the region. We also identified multi-segment models that best represent the observed data.

Lines 687-693: In my opinion, this is rather vague. Please consider making precise recommendations for future research.

We revised it as follows: Future seismic hazard work can be improved by utilizing geophysical data to understand fault structures where strong earthquakes are developing (Xu et al., 2017), applying geodetic data to assess energy accumulation on fault segments (e.g., Yao and Yang, 2023), using microseismicity relocation data to reveal fault asperities (Lay and Nishenko, 2022), and employing dynamic rupture simulations of single and multi-segments to enhance earthquake motion predictions (e.g., Zhang et al., 2017). These studies on fault behaviors, interactions, and multi-segment ruptures are vital for improving seismic hazard assessments. Staying vigilant and proactive in seismic risk management will better protect communities and infrastructure in the NWYR and beyond.

---

## Author Comment (AC4)

The NHESS manuscript "Modeling Seismic Hazard and Landslide Potentials in Northwestern Yunnan, China: Exploring Complex Fault Systems with multi-segment rupturing in a Block Rotational Tectonic Zone" by Cheng et al. focuses on forecasting earthquake activity on the complex northwestern Yunnan fault system.   This paper is generally well written and logically organized.   The authors have broadened the scope of this study by also mentioning implications of their modeling results to ground-motion assessment, regional landslide hazard, and local tectonics. These ancillary topics are treated superficially, but the core modeling methodology is well founded. However, characterization of potential ruptures needs to be broadened and better justified (see Comment 1). Major comments are included below, as well as some minor details that should be easily addressed by the authors.

Major comments:

1. The authors develop four models of multi-segment and multi-fault rupture combinations based on "the segmentation model and fault rupture behaviors", informed largely by historical earthquake ruptures. Given the limited record of finite-rupture observations, this is prone to a great deal of bias [see Stein et al., 2012]. A more objective method is to evaluate all possible segment combinations for a given fault and establish "plausibility filters" (as suggested in Section 4.1) for multifault ruptures [Field et al., 2014]. Then, the results from SHERIFS can be evaluated against the historical record for verification. At minimum, more explanation is needed in Section 2.2 to firmly establish the authors' preferred combinations and perhaps include more possibilities for multi-segment/multi-fault rupture.

Thanks for your comments. I agree with your opinion of the explanation for rupture combinations. For this work, we did not consider the rupture combinations with step width of 5+ km and the strike difference $\geqslant 28°$ between the linked segments. We modified the words in section 2.2 to make the words more reasonable.

2. Uncertainty analysis of the model results is not well described and perhaps incomplete. For example, it is unclear whether uncertainty in fault slip rates, which is detailed in Section 2.1, the regional MFD parameters and the M-A relations are all propagated through to the results.

Thanks for your suggestion. We added the words in section 3.1.

3. In addition, evaluation of the model results is based on NMS ratios, rather than rigorously establishing quantitative prediction errors or goodness-of-fit metrics.

Thank you for your comment. We chose to use NMS ratios for evaluating the model results due to their practical utility in our context. NMS ratios offer a straightforward method to assess model performance relative to a baseline and reflect the goodness-of-fit metrics, as seen in right panel in figure 6. The iteration process focuses predominantly on the fault slip rate, with the remaining portion accounted for by the NMS, thus providing an integrated view of model performance.

4. Description of the PGA calculation is cursory, and it is unclear whether source of uncertainty other than the GMPEs are included.

Thanks for your comment. We added the model limitations in section 4.1.

**4.1 Model Limitations and Mitigation Measures**

Our seismic hazard modeling for NWYR represents our current understanding of average earthquake hazards in the region based on available data. The results are affected by numerous epistemic and aleatory uncertainties inherent in seismic hazard modeling processes, including the MFD, fault geometry, fault type, slip rate, and variability in GMPEs. Mitigating the impact of these uncertainties is critical for accurate seismic hazard assessment.

The MFD relationship, calculated from historical earthquakes, is

essential for determining seismicity rate ratios across different magnitude bins. The deflection of the MFD directly influences the distribution of the modeled seismicity rates. In this study, we chose the G-R relationship over the Y-C relationship due to the regional fragmented tectonic environment. The calculated $b$-value of 0.96 aligns closely with the expected value of 1 found in seismically active regions (Pacheco et al., 1992). To derive earthquake magnitudes on fault segments, we employed rupture scaling relationships based on historical rupture parameters of earthquakes in China as proposed by Cheng et al. (2020), ensuring consistency with unique tectonic characteristics. Achieving more precise MFDs and rupture scaling laws necessitates further refinement in methodology and the use of reliable catalogs specific to the study area.

For fault geometry, type, and slip rates, we relied exclusively on recent field investigation data. In compiling fault rupture models for NWYR, we analyzed these geological data under a unified tectonic stress field, ensuring coordinated fault system movements. The variability in GMPEs is complex, influenced by factors such as earthquake rupture characteristics, seismic wave propagation, and site conditions. Consequently, we incorporated Quaternary sediment site amplification effects on PGA values. Addressing basin effects on ground motion requires dynamic simulations to achieve more precise results.

5. Similarly, uncertainty associated with the landslide hazard analysis is incomplete. See for example, Wang and Rathje [2015]. It is even unclear in this analysis what the parameters of the hazard calculation are (e.g., exposure time, probability model, etc.).

We added the words to explain the landslide hazard analysis as follows:

We used a logistic regression model, well-regarded for its robust performance in machine learning. Unlike previous models (e.g., Nowicki et al., 2014; Wang and Rathje, 2015; Parker et al., 2017) for calculating earthquake-triggered landslide hazards. Our model directly assessed the absolute probability of landslide occurrence, represented as the percentage of the landslide area within a region relative to the total area of the region (Shao et al., 2020). As a result, our hazard estimates have a true probabilistic meaning, reflecting the actual probability of landslide occurrence rather than being merely a formal expression of probability. We then calculated the probabilistic seismic susceptibility for a specific point in time within the study area, which produced a probabilistic PGA distribution map. By using this probabilistic PGA map as input for our model, we can estimate the corresponding probability of earthquake-triggered landslide occurrence. We employed these steps as the basis of our approach to calculating the probability of such landslides.

Minor comments:

(6) L20: Specify "ductile flow of the lower crust" to be clearer.

Thanks! We revised it to "ductile flow of the lower crust with low shear-wave velocity".

(7) L32 and throughout: "averagely" -> "on average".

Revised.

(8) L65: Is the "low velocity belt" delineated by the faults located in the lower (i.e., ductile region) or upper crust (i.e., the host rock of the faults)?

Revised. We mean that the low velocity belt with lower-crust flow

(9) L106: Unclear what the "pre-earthquake period" refers to.

Revised.

(10) L108, 112: "errors"-> I think you mean "uncertainty".

Revised.

(11) L151-153: Indicate some brief description of GMPEs and site conditions used, as this is key to PGA estimates.

Thanks to your advice. We added the words for GMPEs and site conditions in Line 153 and Line 157.

(12) L193: Reference Figure 2.

Thanks! We added it in Line 199.

(13) L303 and throughout this section: "integrated"->"included" or similar.

We revised it in Line 303.

(14) L313-316: This seems like conjecture.   Any evidence to support this inference?

The fieldwork in this region is relatively scare, as the rugged and uneven terrain. Here, we modified the words from "hinder" to "strongly impacted on" in Line 342.

(15) L336 and throughout this section: Need to distinguish the regional MFD (input to model) from the on-fault MFD (output).

Thanks! We added the words in Line 378 and Line 395.

(16) L357, 434: The Wells and Coppersmith (1995) relations are dated at this point. Better to use, for example Leonard [2010], or a similar recent

study as an alternative to Cheng et al. (2020). See summary by Stirling et al. [2013].

We selected the scaling relationship of Cheng et al. (2020) because it is specifically developed for earthquakes in mainland China, making it more regionally appropriate for our study. By comparing this with the well-established scaling relationship of Wells and Coppersmith (1994), which is based on a global dataset of both interplate and intraplate earthquakes, we aim to assess whether regional-specific models offer improved accuracy over more generalized, globally applicable models.

The words to explain the reasons in the context are as follows: For the rupture scaling relationships, most of them are developed for plate boundary regions (Stirling et al., 2013). In this study, we selected a regression scaling relationship based on a dataset of earthquakes from mainland China (Cheng et al., 2020) and compared the results with the widely used rupture scaling relationship of Wells and Coppersmith (1994), which incorporates global data from both interplate and intraplate earthquakes.

(17) L406: Shouldn't some goodness-of-fit metric be used then?

Thank you for your insightful suggestion. We agree that incorporating a quantitative goodness-of-fit (GOF) metric would provide a more rigorous assessment of model performance. However, in this study, the NMS ratios not only reflect the regression fit but also clearly indicate which segments have lower NMS ratios. This makes it easier to identify which segments deviate from the modeled seismicity rates, providing valuable insights for comparison. Nonetheless, we will consider adding a GOF metric to complement the NMS ratios and provide a more quantitative evaluation.

(18) L448: There hasn't been any explanation on how these prediction intervals are calculated. Please include a detailed description, particularly which sources of uncertainty this pertains to.

Thank you for your insightful comment regarding the calculation of prediction intervals. We added the words in the 3$^{rd}$ paragraph in section 3.1, Methodology.

"In these steps, the b-value from historical earthquakes, the rupture scaling law of the faults, and the fault slip rates are typically accompanied by significant uncertainties. SHERIFS used the random sampling method to explore the uncertainty bounds. The rates are derived while examining uncertainties related to earthquake magnitudes, the duration of the completeness period, and the low number of observed earthquakes for

larger magnitudes, using a Monte Carlo approach (Chartier et al., 2021).
For each branch of the logic tree in the random sampling, it generates a
corresponding number of models that match the total count of random
samples. For each model, the slip-rate value is selected uniformly within
its uncertainty bounds, scaling law parameters are chosen independently
from a Gaussian distribution within their error bounds, and the b-value is
picked from the user-defined range. All these uncertainties propagate to
the final step of calculating seismicity rates with uncertainties. ”

(19) L463: "branches" of what?

Thanks! We revised it to "branches of GMPEs".

(20) Figures: Font size is very small, to the point where the labels and
numbers are unreadable.

Thanks! We revised the labels and numbers to be readable.

References cited in review

Field, E. H., et al. (2014), Uniform California Earthquake Rupture
Forecast, version 3 (UCERF3)--The time-independent model, Bull.
Seismol. Soc. Am., 104, 1122-1180.

Leonard, M. (2010), Earthquake fault scaling: Self-consistent relating of rupture length, width, average displacement, and moment release, Bull. Seismol. Soc. Am., 100(5A), 1971-1988.

Stein, S., R. J. Geller, and M. Liu (2012), Why earthquake hazard maps often fail and what to do about it, Tectonophys., 562-563, 1-25, doi:https://doi.org/10.1016/j.tecto.2012.06.047.

Stirling, M., T. Goded, K. Berryman, and N. Litchfield (2013), Selection of Earthquake Scaling Relationships for Seismic-Hazard Analysis, Bull. Seismol. Soc. Am., 103(6), 2993-3011, doi:10.1785/0120130052.

Wang, Y., and E. M. Rathje (2015), Probabilistic seismic landslide hazard maps including epistemic uncertainty, Eng. Geol., 196, 313-324, doi:https://doi.org/10.1016/j.enggeo.2015.08.001.